# Prominent members of the human gut microbiota express endo-acting O-glycanases to initiate mucin breakdown

Lucy I. Crouch [1,12 ✉], Marcelo V. Liberato[2], Paulina A. Urbanowicz [3], Arnaud Baslé[1], Christopher A. Lamb [4], Christopher J. Stewart[5], Katie Cooke[5], Mary Doona[4], Stephanie Needham[6], Richard R. Brady[7], Janet E. Berrington[8], Katarina Madunic [9], Manfred Wuhrer [9], Peter Chater[1], Jeffery P. Pearson[1], Robert Glowacki[10], Eric C. Martens [10], Fuming Zhang[11], Robert J. Linhardt [11], Daniel I. R. Spencer[3] & David N. Bolam [1 ✉]

The thick mucus layer of the gut provides a barrier to infiltration of the underlying epithelia by both the normal microbiota and enteric pathogens. Some members of the microbiota utilise mucin glycoproteins as a nutrient source, but a detailed understanding of the mechanisms used to breakdown these complex macromolecules is lacking. Here we describe the discovery and characterisation of endo-acting enzymes from prominent mucin-degrading bacteria that target the polyLacNAc structures within oligosaccharide side chains of both animal and human mucins. These O-glycanases are part of the large and diverse glycoside hydrolase 16 (GH16) family and are often lipoproteins, indicating that they are surface located and thus likely involved in the initial step in mucin breakdown. These data provide a significant advance in our knowledge of the mechanism of mucin breakdown by the normal microbiota. Furthermore, we also demonstrate the potential use of these enzymes as tools to explore changes in O-glycan structure in a number of intestinal disease states.

[1] Biosciences Institute, Faculty of Medical Sciences, Newcastle University, Newcastle upon Tyne, UK. [2] Universidade de Sorocaba, Programa de Processos Tecnológicos e Ambientais, Sorocaba, Brasil. [3] Ludger Ltd, Culham Science Centre, Abingdon, UK. [4] Department of Gastroenterology, Newcastle upon Tyne Hospitals NHS Foundation Trust, Newcastle upon Tyne, UK. [5] Translational and Clinical Research Institute, Faculty of Medical Sciences, Newcastle University, Newcastle upon Tyne, UK. [6] Department of Histopathology, Newcastle upon Tyne Hospitals NHS Foundation Trust, Newcastle upon Tyne, UK. [7] Department of Colorectal Surgery, Newcastle upon Tyne Hospitals NHS Foundation Trust, Newcastle upon Tyne, UK. [8] Newcastle Neonatal Service, Royal Victoria Infirmary, Newcastle upon Tyne, UK. [9] Centre for Proteomics and Metabolomics, Leiden University Medical Centre, Leiden, Netherlands. [10] Department of Microbiology and Immunology, University of Michigan Medical School, Ann Arbor, MI, USA. [11] Department of Chemistry and Chemical Biology, Centre for Biotechnology and Interdisciplinary Studies, Rensselaer Polytechnic Institute, Troy, NY 12180, USA. [12] Present address: Institute of Microbiology and Infection, School of Biosciences, University of Birmingham, Birmingham B15 2TT, UK. ✉email: l.i.crouch@bham.ac.uk; david.bolam@ncl.ac.uk

The human gastrointestinal (GI) tract is home to a large and complex community of microbes known as the human gut microbiota (HGM), with the greatest densities assembling in the large intestine were numbers of bacterial cells are estimated to be ~100 trillion[1]. The mucus layer shields the host epithelial cells of the GI tract from both the normal microbiota and enteric pathogens. Mucus is predominantly composed of gel-forming mucins, which are complex glycoproteins secreted by the epithelial cells[2]. Different mucin genes are expressed in different mucosal surfaces throughout the body and mucins are at least 50% O-glycan by mass[3]. In the colon, MUC2 is the most abundant gel-forming mucin and is composed of ~80% glycan[1]. While the number of different monosaccharides and types of sulphate decoration making up mucin oligosaccharide side chains are

relatively limited, the order in which they can be assembled is hugely variable (Fig. 1a). The heterogeneity between individual O-glycan chains leads to a highly complex macromolecule and it is this complexity that provides some resistance to microbial degradation and contributes to the mucus layers' protective role[4]. Despite this heterogeneity, some prominent bacterial members of the microbiota have developed the capacity to graze on mucins, including certain *Bacteroides* spp. and *Akkermanisa muciniphila*[5–9]. This ability is thought to be critical to the initial colonisation by the microbiota in a new-born and therefore to the development of a healthy adult microbiota[10]. Mucin grazing also enables survival during the absence of diet-derived glycans[11] and non-mucin degrading species have been shown to be cross-fed by mucin degraders, contributing to the long-term survival and

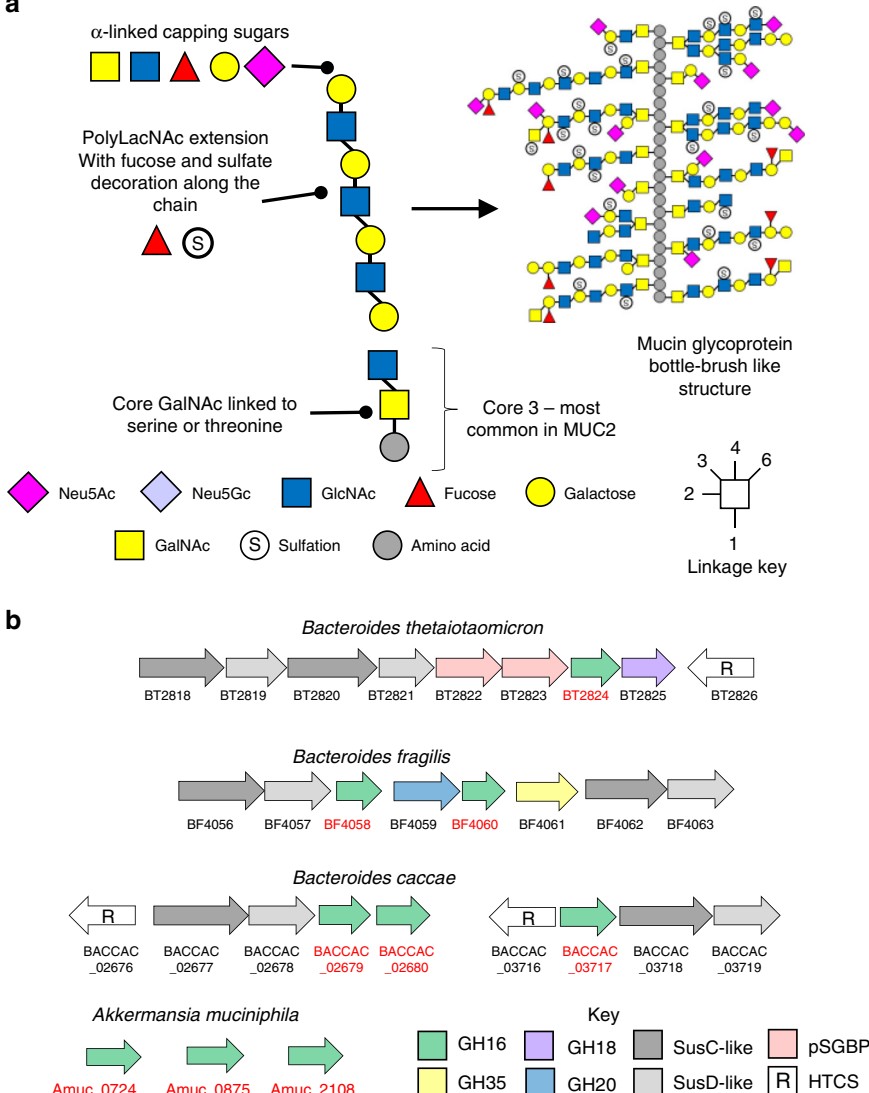

**Fig. 1 Mucin structure and genomic context of the loci encoding the mucin-associated GH16 enzymes. a** Left: the main structural features of a model mucin O-glycan chain. All mucin oligosaccharides are linked via an α-GalNAc to serine and threonine residues in the peptide backbone. A number of different core structures are then attached, with core 3 (shown) being the most common in the large intestinal MUC2. The cores are then often extended with polyLacNAc repeats of varying lengths which are decorated along their length by sulfation and fucosylation and capped at the non-reducing end by a variety of α-linked monosaccharides. Right: a model of an intestinal mucin glycoprotein showing complexity and variability of glycan chains attached to peptide backbone. **b** Genetic context of the GH16 encoding genes identified as being upregulated in the four species shown during growth on mucin (see Supplementary Figs. 1–3). In *Bacteroides* spp. the GH16 genes (highlighted red) are part of discrete polysaccharide utilisation loci (PULs), cluster of co-regulated genes encoding glycan degradation and uptake apparatus (SusC-like and SusD-like outer membrane proteins, additional CAZymes and putative surface glycan binding proteins (pSGBPs), often adjacent to a hybrid two component system (HTCS) sensor-regulators that likely control expression of the associated PUL. Glycan utilisation genes are not organised into PULs in the *A. muciniphila* genome.

stability of the microbiota[12,13]. In contrast, aberrant or excess degradation of the mucosal layer by the normal microbiota has been linked to enhanced pathogen susceptibility, inflammatory bowel disease (IBD) and even colorectal cancer[11,14].

Despite the importance to gut health of mucin breakdown by the microbiota, little is known about the molecular details of this process. Current models of mucin degradation propose extracellular sequential trimming of terminal sugars from the O-glycan side chains by extracellular exo-acting glycosidases to eventually expose the peptide backbone for proteolysis[15]. However, this extracellular 'exo-trimming' model is based only on the activity of currently characterised mucin active enzymes, such as sialidases and fucosidases. The exo-trimming model is also in direct contrast with the degradation pathways used by Gram-negative Bacteroidetes, known as the Sus paradigm[16]. Sus-like systems derive their name (Starch utilisation system) after the first such system characterised, but each Sus-like apparatus targets a distinct glycan, and many *Bacteroides* spp. contain tens to hundreds of these systems. In general, in Sus-like systems, a surface endo-acting glycanase cleaves the substrate (polysaccharide or glyco-conjugate) into smaller oligosaccharides for uptake by SusC/D outer membrane complexes[17–19].

Here we describe the discovery and characterisation of endo-acting glycoside hydrolases expressed by mucin-degrading members of the HGM that are able to cleave the O-glycan chains of a range of different animal and human mucins. These O-glycanases display endo-β1,4-galactosidase activity and specifically target the polyLacNAc structures that comprise the main backbone of many mucin glycan chains. Furthermore, many of the enzymes are predicted to be surface located and thus support a model where the initial steps of mucin degradation by gut bacteria involves the extracellular removal of oligosaccharides from the glycoprotein. We also provide evidence these endo O-glycanases could be exploited as tools to explore the composition of human O-glycans from a range of different sources, with potential applications in both basic research and precision medicine.

## Results

**Identification of GH16 enzymes expressed during growth on mucin.** The foundation to this work was the analysis of previously published transcriptomic and proteomic data available from four prominent mucin degrading HGM species (*Bacteroides thetaiotaomicron*, *B. fragilis*, *B. caccae* and *A. muciniphila*) grown on intestinal mucins[8,11,20–23]. The genes and proteins highlighted in these studies included many putative exo-acting enzymes from CAZy families (carbohydrate active enzymes; CAZymes) that have previously been identified as involved degradation of O-glycans, such as sialidases (GH33) and fucosidases (GH29 and GH95; Supplementary Figs. 1–3). Surprisingly, some of the most upregulated CAZymes in all species analysed were from glycoside hydrolase family 16 (GH16). This was unexpected as GH16 enzymes have been predominantly characterised as targeting a variety of marine or terrestrial plant polysaccharides, specifically β1,3 or 1,4 glycosidic bonds of glucans and galactans almost exclusively in an endo-acting fashion (Supplementary Fig. 4). For *Bacteroides* species belonging to the HGM, the GH16 enzymes that have been characterised so far include specificities towards porphyran, agarose, β-1,3-glucan, and mixed linkage glucan[24–27]. The genetic loci encoding the GH16 enzymes upregulated on mucin also contain genes encoding SusCD pairs for the *Bacteroides* spp. and, in the case of *B. fragilis*, other predicted CAZymes (Fig. 1b). The CAZymes encoded in the *A. muciniphila* genome are not organised into loci. The modular structure of the mucin-associated GH16 enzymes are shown (Fig. 2).

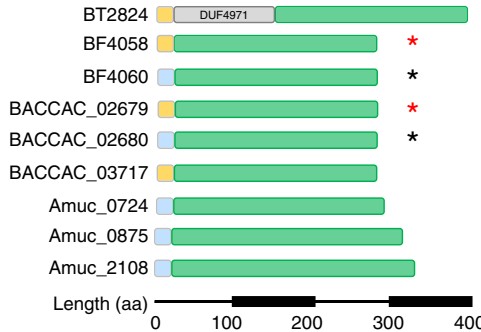

**Fig. 2 Domain structure of the mucin-associated GH16 enzymes characterised in this study.** GH16 catalytic domain (green), Type I signal peptide (light blue), Type II signal peptide (orange). BT2824 also has an N-terminal DUF4971 domain (PF16341) of unknown function. Sequence identity is between 22 and 38% for most of the enzymes, but two pairs of the GH16 enzymes are close homologues - BF4058 and BACCAC_02679 (red asterisk) display 87% identity, while BF4060 and BACCAC_02680 (black asterisk) display 79% identity.

Specifically, the GH16 enzymes identified are a part of subfamily 3, which is a large and sequence-diverse subfamily characterised predominantly as β1,3/4-glucosidases found in Metazoa, Fungi, Archaea and Bacteria[28]. A total of nine GH16 enzymes were identified from the four species (Figs. 1b, 2, and Supplementary Figs. 1–3). Five of the nine GH16 enzymes are predicted lipoproteins and therefore likely cell surface associated (Supplementary Table 1). Interestingly, these GH16 family members generally had a relatively low sequence identity between 24–34%. The exceptions to this were two pairs of *B. fragilis* and *B. caccae* enzymes: BF4058 and BACCAC_02679 and BF4060 and BACCAC_02680 with 87 and 79% identity, respectively (Fig. 2 and Supplementary Table 2).

**Phylogenetic analysis of mucin-associated GH16 enzymes.** The protein sequences of the nine GH16 family members identified as upregulated during growth on mucins were compared with the characterised GH16 family members from the CAZy database (Supplementary Fig. 5 and Supplementary Table 3). The phylogenetic tree indicates that the mucin-associated GH16 enzymes are most closely related to the β-glucanase GH16 family members rather than those with activities on β-galactans, xyloglucan or chitin- β1,6-gluconotransferases. Another analysis of the GH16 subfamily 3 protein sequences indicate that seven of the mucin-associated GH16 enzymes cluster together (Supplementary Fig. 6). The branch where they sit is composed of proteins only from mucosal-associated organisms, including known pathogens. Two of the *A. muciniphila* enzymes (Amuc_0724 and Amuc_0875) cluster in a different branch and are relatively close to GH16 family members that have been characterised as having endo β-1,3-galactanase activity (Supplementary Fig. 6). The two separate clusters indicate that the mucin-associated GH16 enzymes have evolved twice from different β-glucanase ancestors (Supplementary Fig. 6). The enzymes that cluster with these sequences potentially have similar activities. While the mucin-associated GH16 enzymes are present in prominent mucin degrading Gram-negative bacteria such as *Bacteroides* spp., inspection of the genomes of mucin-degrading Gram positive members of the gut microbiota, including some *Ruminococcus* and *Bifidobacteria* spp., revealed no evidence for these enzymes[29].

**The mucin-associated GH16 genes encode endo-acting O-glycanases.** To explore the activity of the nine mucin-upregulated GH16 family members, the recombinant forms of the enzymes

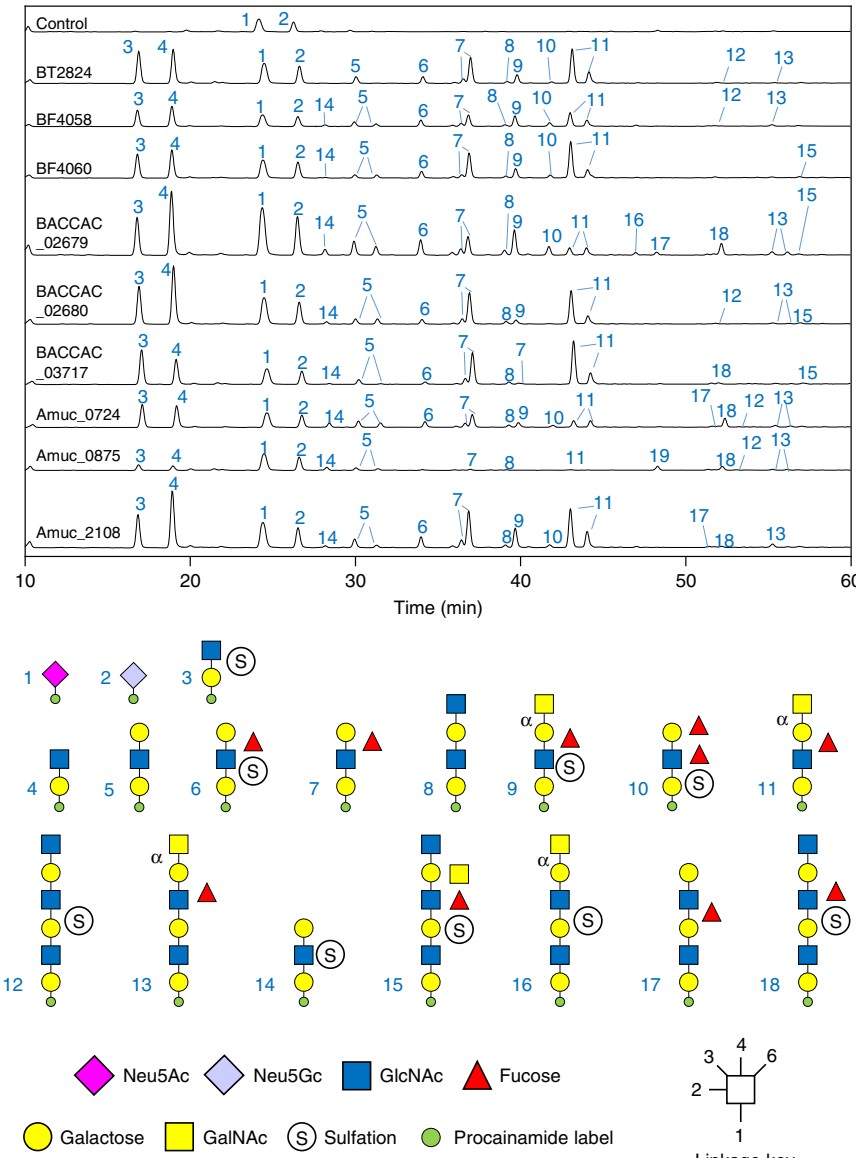

**Fig. 3 Activity of the GH16 O-glycanases against porcine small intestinal mucin.** SI mucin was incubated with the GH16 enzymes alongside a broad-acting sialidase (BT0455^GH33). The control is sialidase-only. Products of small intestinal mucin digestion were labelled with procainamide at the reducing end and analysed by LC-FLD-ESI-MS. The results show a variety of oligosaccharides are released by the GH16 enzymes and majority are similar between the samples. The oligosaccharides have a variety of fucose and sulphate decorations. Species capped with α-GalNAc were determined using exo-acting enzymes specific to that sugar and linkage (see Supplementary Fig. 8).

were screened against porcine small intestinal (SI) mucin and porcine gastric mucins (PGM type II and III; Supplementary Fig. 7). Initial analysis by thin layer chromatography (TLC) suggested that all nine enzymes were active against both SI and gastric mucins and released a range of products from these glycoproteins that are larger than monosaccharides, suggesting endo-like cleavage of the O-glycan chains.

To investigate the identity of these products in more detail, the glycans were labelled at their reducing end with the fluorophore procainamide and analysed by liquid chromatography-fluorescence-detection-electrospray-mass spectrometry (LC-FLD-ESI-MS) and the glycan structures determined by MS/MS (Fig. 3). The data show that all the GH16 enzymes produce oligosaccharides of alternating hexose and HexNAc sugars with a variety of lengths. The reducing ends were all hexoses, indicating hydrolysis occurred at β-galactose (α-galactose only occurs in mucins as a terminal sugar in blood group B structures and GH16 enzymes

only target β-linked sugars) and the products also had a range of fucose and sulphate decorations, revealing these can be accommodated by the GH16 enzymes. Overall, these data indicate that the nine GH16 enzymes are all endo-acting β-galactosidases that are active on the O-glycan side chains of mucin (O-glycanases).

Notably, sialic acid (SA) was never observed as a decoration on any products released by the enzymes, even though SA is present on mucin glycans (Fig. 3). These data suggest this terminal sugar decoration cannot be accommodated by the GH16 O-glycanases and, as a result, the broad acting sialidase BT0455^GH33 was included in all assays to maximise access of the GH16 enzymes to the mucin chains.

To investigate the composition of the oligosaccharides that are released by the GH16 O-glycanases in greater detail we digested the GH16 products with a series of exo-acting glycosidases of known specificity. For example, digestion with glycosidases

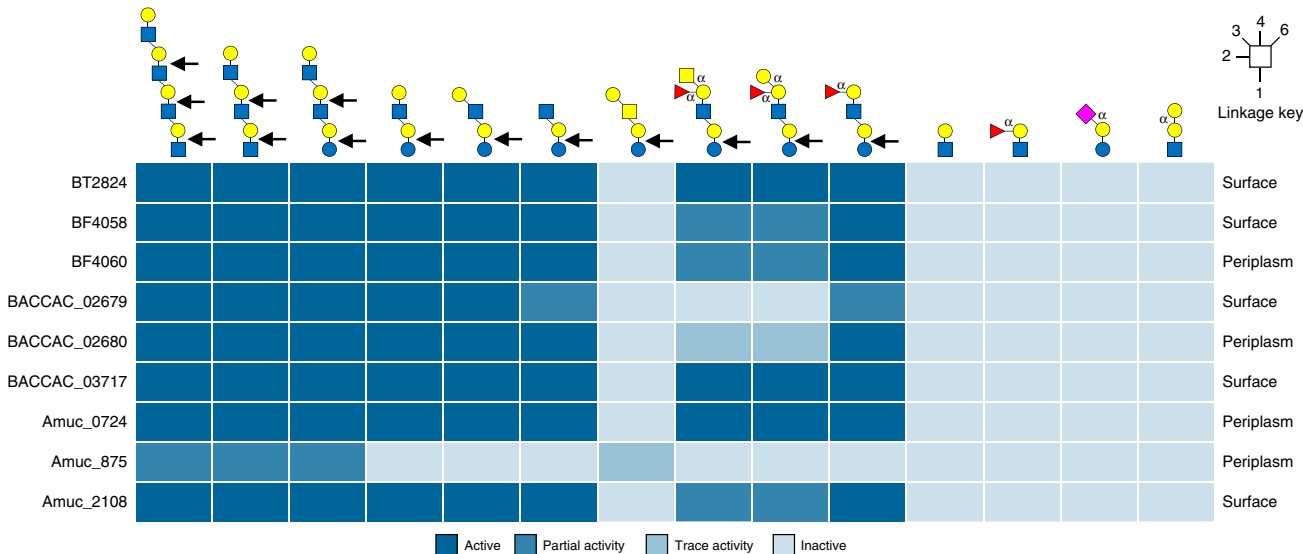

**Fig. 4 Heat map showing the activity of the GH16 O-glycanases against different oligosaccharides.** The data summarises the specificity of the GH16 O-glycanases described in this report. From left to right the glycans are TetraLacNAc, TriLacNAc, paraLacto-N-neohexaose, Lacto-N-neotetraose, Lacto-N-tetraose, Lacto-N-triose, Galβ1,3GalNAc β1,3Galβ1,4Glc tetrasaccharide Blood group A hexasaccharide, Blood group B hexasaccharide, Blood group H pentasaccharide, LacNAc, Blood group H tetrasaccharide II, 3-sialyllactose and P1 antigen. The linkages are β unless otherwise labelled and the bonds cleaved are indicated by the black arrows. Partial and trace activity are the estimation of greater than or less than 50% degradation, respectively, under the assay conditions used. A more detailed summary can be found in Supplementary Table 4. The predicted cellular locations of each enzyme is indicated on the far right of each row.

specific to either the α-GalNAc or α-galactose found on blood group A or B structures, respectively, enabled identification of these non-reducing end decorations on some GH16 products (Supplementary Fig. 8). The use of endo- and exo-acting enzymes in combination against different mucins could be a powerful tool in exploring different structures in O-glycan research and identification of disease biomarkers.

**Investigating the specificity of the GH16 O-glycanases.** A range of defined oligosaccharides were used to further probe the specificity of the O-glycan active GH16 enzymes (Supplementary Figs. 9–12). TriLacNAc represents the repeating unit of O-glycans side chains and was hydrolysed at two of the Galβ1,4GlcNAc linkages by the GH16 mucinases. The middle bond is the first to be cleaved to produce two trisaccharides, one of these is then hydrolysed further to produce GlcNAc and GlcNAcβ1,3Gal. This order was determined by identifying the products using digestion with specific exo-acting enzymes (Supplementary Fig. 9). The activity against TriLacNAc revealed that all nine GH16 enzymes are endo β1,4-galactosidases with a requirement for a β1,3-linked sugar at the -2 position.

The GH16 mucinases were also tested against other defined oligosaccharides to explore differences between the enzymes and the importance of different subsites in substrate recognition (Supplementary Figs. 10–12). In summary, for the positive subsites, all displayed a preference for O-glycans over milk oligosaccharides (which are built on a lactose core), indicating that a GlcNAc is preferred at the +1 site rather than a Glc. In terms of the negative subsites, blood group sugars in the -3' (fucose) and -4 (GalNAc or Gal) subsites are tolerated in most cases but reduce the rate of hydrolysis (Supplementary Table 4). A summary of the specificities of the GH16 O-glycanases described in this report is shown in Fig. 4.

**Endo O-glycanase activity on the cell surface.** To assess if there is endo O-glycanase activity on the cell surface we used whole cell assays (Fig. 5 and Supplementary Fig. 13). Bacterial cultures

grown on PGM III were harvested, washed and exposed to either TriLacNAc or TetraLacNAc. The data revealed the same pattern of degradation as that produced by the recombinant GH16 O-glycanases against TriLacNAc. The identity of the products was confirmed using diagnostic assays using exo-acting enzymes of known specificity for two of the time points (Fig. 5b and Supplementary Fig. 13). These data indicate that GH16 O-glycanase-like activity is present on the surface of all four of the bacterial species studied here.

**Activity against polyLacNAc structures in non-mucin host glycan.** Keratan sulphate (KS) chains are anchored to the protein through N-linkages, O-linkages and O-mannosylation are termed KS-I, KS-II and KS-III, respectively[30] (Supplementary Fig. 4). Examples of areas of the body enriched in KS-I, KS-II and KS-III include the cornea, skeletal and brain, respectively[30]. Keratan sulphate is present in the GI tract from sloughed off epithelial cells and also dietary sources.

Keratan sulphate has a similar structure to O-glycans and is also composed of a repeating polyLacNAc structure with 6 S sulfation possible on both the galactose and GlcNAc, but with less fucosylation and sialylation than most mucins. The nine GH16 O-glycanases were found to be active against both egg and bovine corneal keratan sulphate and the released products indicate that a significant number of sulphate groups can be tolerated by the enzymes (Supplementary Fig. 14). Degradation of KS also supports the previous finding that fucose decorations are not required for activity of the GH16 O-glycanases. Activity against this KS substrate also demonstrates that the O-glycanases target polyLacNAc chains in a range of glycans, likely including keratan sulphate in the gut, although we could not test growth on KS as the glycan was only available in small amounts.

**Activity of the GH16 O-glycanases against classical GH16 substrates.** The activity of GH16 enzymes against host glycans is unusual in comparison to the specificities displayed for previously characterised members of the family. The activities of the GH16

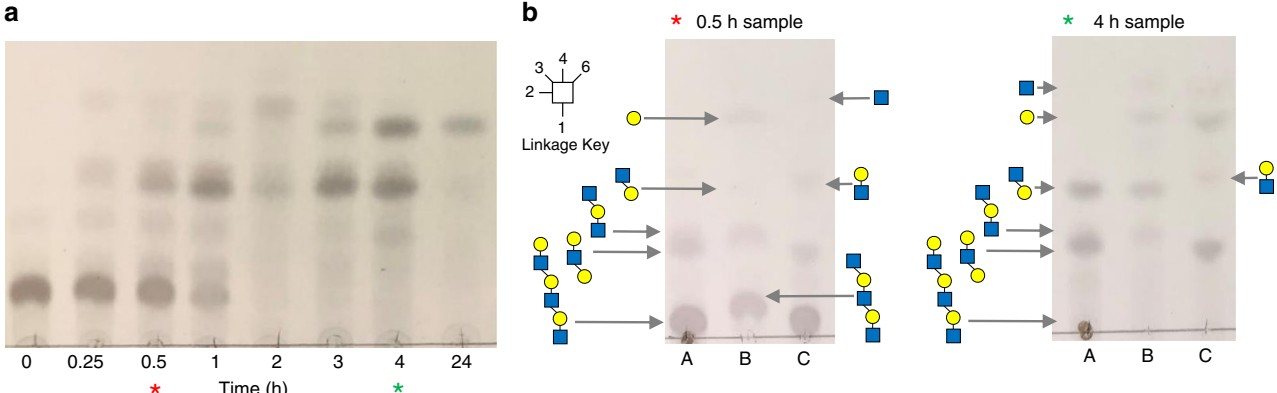

**Fig. 5 Surface activity of mucin-grown *A. muciniphila* against TriLacNAc. a** *A. muciniphila* cells were grown on PGM III, harvested and TriLacNAc added to assess surface enzyme activity. Samples were analysed at different time points following addition of TriLacNAc. Equivalent data for the *Bacteroides* species are shown in Supplementary Fig. 13. **b** Samples from two different time points of the *A. muciniphila* whole cell assay were incubated with two different exo-acting enzymes of known specificity to identify the products. The asterisks indicate the time points analysed. The top and bottom of the glycan structures shown is the non-reducing and reducing end, respectively. A: original sample, B: +β1,4-galactosidase BT0461[GH2], C: +broad-acting β-GlcNAc'ase BT0459[GH20]. For example, in both time points, the two different types of trisaccharides can be seen to be hydrolysed in the separate digests: (1) Galβ1,4GlcNAcβ1,3Gal disappears in 'B' and products Gal and GlcNAcβ1,3Gal are now present, (2) GlcNAcβ1,3Galβ1,4GlcNAc disappears in 'C' and products GlcNAc and Galβ1,4GlcNAc are now present. These experiments were carried out once, but multiple pilot experiments were run and were consistent with the data shown. The source data are provided in the source data file.

O-glycanases were tested against polysaccharides previously shown to be substrates for GH16 family members to assess their level of O-glycan specificity (Supplementary Fig. 15). No activity could be detected for any of the enzymes against agarose, κ-carrageenan, porphyran, pectic galactan, xyloglucan or chitin. However, Amuc_0724 displayed significant endo-like activity against laminarin and weak activity against barley β-glucan and lichenan. BF4060, BACCAC_02680 and BACCAC_03717 also displayed some very weak activity against laminarin. The possible rationale for the activity of Amuc_0724 against Glc configured substrates is discussed below. Other non-mucin host poly-saccharides are also present in significant amounts in mucosal surfaces, including chondroitin sulphate (CS), heparin (Hep) and hyaluronic acid (HA). The O-glycan active GH16 enzymes were also tested against these polysaccharides and no significant activity could be found, except for a small amount of low molecular weight product released from Hep (Supplementary Fig. 15).

Overall these data reveal that the nine GH16 enzymes analysed are endo acting β1,4-galactosidases that display a preference for polyLacNAc structures found in mucins and other similar glycans such as KS.

**Crystal structures of O-glycan active GH16 family members.** To investigate the structural basis for O-glycan specificity displayed by the GH16 O-glycanases we attempted to obtain crystal structures of these enzymes in complex with substrate. Crystal trials were set up with both wild-type protein and mutants of the catalytic nucleophile. We also co-crystallised the different proteins with TriLacNAc to obtain substrate or product complexes where possible. Six separate data sets were collected from four of the enzymes. The apo structures of BACCAC_02680, BACCAC_02680[E143Q], BACCAC_03717 and Amuc_0724 were obtained to 2.0, 2.1, 2.1 and 2.7 Å, respectively. Structures of BF4060 and BACCAC_02680[E143Q] were also obtained with the Galβ1,4GlcNAcβ1,3 Gal product present in the negative subsites (despite the latter enzyme being a catalytic mutant; Supplementary Fig. 16) to 3.3 and 2.0 Å (Fig. 6, Supplementary Tables 5–7, and Supplementary Figs. 16–20). The electron density of the trisaccharide product allowed us to model in the sugars, the conformations were checked using Privateer, and these conformations correlated

with what had been seen previously in other GH16 structures (Supplementary Fig. 17 and Supplementary Table 7)[31,32].

All of the GH16 enzymes comprise a β-jellyroll fold, characteristic of the family, consisting of two β-sheets composed of β-strands that form the core fold, which were superimposable with other GH16 structures previously published. A cleft running along the concave surface of the enzymes contains the active site and where the trisaccharide product was bound in the cases of BF4060 and BACCAC_02680[E143Q] (Fig. 6a). While the location of the substrate binding site is conserved in the GH16 family, the structures of these clefts vary depending on substrate specificity (Supplementary Fig. 18a). Some form a tight tunnel for linear undecorated glycans (e.g., agarase from *Zobellia galactanivorans*[33]), others are much more open to accommodate decorations (e.g., xyloglucanase from *Tropaeolum majus*[34]), while some GH16 enzymes have substrate binding clefts that are curved to optimise binding to highly curved glycans such as laminarin[35]. There is also a single example of a GH16 family member that has evolved a pocket-like active site to recognise and cleave a specific disaccharide from the terminus of glycan chains in gastric mucin[36] (Supplementary Fig. 18b). The key structural features that modulate the shape of different clefts are the surrounding loops and short α-helices extending from the β-strands of the core fold (exemplified in Supplementary Fig. 19). These extensions have been likened to fingers that interact with substrate, thereby dictating specificity, and that nomenclature is used herein[37]. BF4060 and BACCAC_02680[E143Q] have four fingers and BACCAC_03717 and Amuc_0724 have five out of six possible fingers that have been observed previously in other GH16 structures[37].

Inspection of the BACCAC_02680[E143Q] and BF4060 structures with product reveal most of the interactions between enzyme and sugar are with the Gal at −1 and GlcNAc at −2. The −1 subsite in BACCAC_02680[E143Q] is composed of a number of aromatics, which are also a common feature of the GH16 structures available (Fig. 6b). This enzyme possesses four fingers (numbers 1, 3, 5 and 6) that extend towards the cleft, with fingers 1 and 3 sandwiching the negative subsites and fingers 5 and 6 sandwiching the positive subsites. Finger 3 contains the sequence motif for GH16 subfamily 3, which consists of three

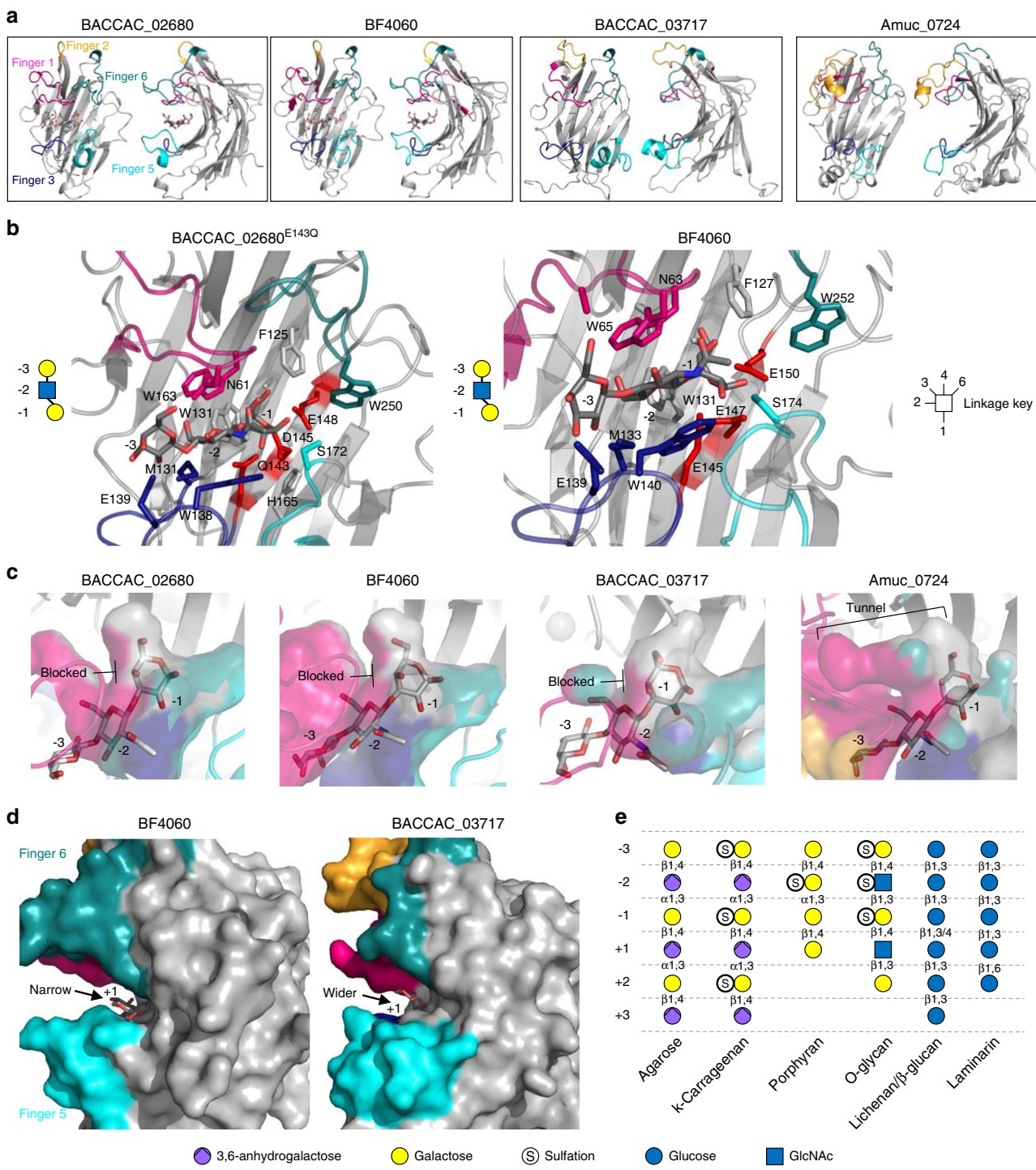

tryptophans interspaced by other residues[28]. BF4060 and BACCAC_02680 display 79% identity and unsurprisingly the structures of these two enzymes are almost identical in the cleft region. In contrast, BACCAC_03717 and Amuc_0724 both possess a finger 2 (in addition to 1, 3, 5 and 6) and this has a more variable topology than the other fingers (Fig. 6a). For Amuc_0724, finger 2 sits over the top of finger 1, but in the BACCAC_03717 structure it points away from the cleft. This could reflect the flexibility of finger 2 in this enzyme and could potentially come down over loop 1 in solution like in the Amuc_0724 structure. The B-factor putty projections of the GH16 crystal structures show finger 2 is dynamic in the BACCAC_03717 (Supplementary Fig. 19) and alternative

conformations of individual fingers from of other GH16 family members has been observed previously, a finding which is indicative of flexibility[38].

**Structural basis for specificity of the GH16 O-glycanases**. Previously characterised GH16 family enzymes target a variety of β-glucan and galactan substrates (Supplementary Fig. 4). Glucose and galactose differ only in the hydroxyl group at C4 being equatorial or axial, respectively. Anhydrogalactose is also present in agarose and carrageenan and sulfation in porphyran and carrageenan (Fig. 6e). Porphyran and carrageenan are 6 S and 4 S sulfated, respectively, and these decorations would therefore point into the GH16 binding cleft at subsites −2 and −1, respectively.

**Fig. 6 Structures of four of the O-glycan active GH16 family members characterised in this study. a** Crystal structures of BACCAC_02680[E143Q], BF4060, BACCAC_03717, and Amuc_0724. The loops extending from the active site that are proposed to be involved in substrate specificity in GH16 enzymes are termed 'fingers' and are colour coded. **b** Subsites −1 to −3 of BACCAC_02680[E143Q] and BF4060 have the product of TriLacNAc cleavage bound (Galβ1,4GlcNAcβ1,3 Gal; shown in symbol form next to the structures with the sugar in each subsite labelled). The residues interacting directly with sugar are shown as sticks. The aromatic residues shared with β-glucanase GH16 family members that drive specificity for a β1,3 between the −1 and −2 sugars are shown (W129, W138 and W131, W140; See Supplementary Fig. 20 for active sites of BACCAC_03717, and Amuc_0724). **c** A surface representation of the regions surrounding the −1 subsite showing the selection for the axial O4 of Gal in the three *Bacteroides* enzymes, while Amuc_0724 has a more open 'tunnel' like space that appears to also allow accommodation of the equatorial O4 of Glc. The product from BACCAC_02680[E143Q] was overlaid in the BACCAC_03717, and Amuc_0724 structures. Colours represent the different 'fingers'. **d** A view of the predicted +1 subsite of BF4060 and BACCAC_03717 overlaid with the glucose from the +1 subsite of a laminarinase from *Phaenerochaete chrysosporium*. The +1 subsites are much more closed for BF4060 and BACCAC_02680[E143Q] compared to BACCAC_03717, and Amuc_0724. **e**, An overview of the monosaccharides occupying the different subsites in GH16 family members with different activities. Linkages also shown. It should be noted that the sulfation will be variable along the O-glycan chain and there will also be fucose decorations. This situation is similar to porphyran, where the polysaccharide can have a variable composition, but the subsite occupancy shown here reflects the observations of structures of enzyme-glycan complexes currently available for porphyranases.

Structural features characteristic to O-glycans include alternating Glc and Gal configured sugars and additionally the presence of GlcNAc, which is not found in other GH16 substrates (Fig. 6e). Furthermore, 6 S is found on both Gal and GlcNAc and 3 S is possible on the galactose at the non-reducing ends of O-glycan chains.

The new GH16 structures reported here were systematically compared to the GH16 structures available for glucanases, laminarinases, porphyranases, carrageenases, xyloglucanases and other activities. Each subsite was analysed in comparison to the structures of GH16 enzymes with other specificities to understand the structural basis for O-glycan specificity. This detailed analysis highlighted four structural features of the mucin active GH16 family members that tailor these enzymes towards O-glycans.

Firstly, in the −1 subsites of the structures from *Bacteroides* spp., the closed space around the O4 hydroxyl explains why only Gal configured sugars can be recognised as the equatorial O4 of glucose would not be accommodated (Fig. 6c). The structure of Amuc_0724 in this area is much more open and is a likely explanation for this enzymes additional activity against laminarin (Supplementary Fig. 15). Furthermore, the open space at the O4 in Amuc_0724 is a potential pocket for sulfation that the *Bacteroides* spp. enzymes would not be able to accommodate (Supplementary Fig. 20). Phylogenetic analysis reveals the mucin active GH16 enzymes are likely to have derived from β-glucanases rather than β-galactanases (Supplementary Fig. 5), however in the −1 subsite the selection is for galactose rather than glucose. This finding indicates the enzymes have evolved specificity for O-glycan structures from a β-glucanases ancestor (s). For the GH16 family as a whole, there is no conserved way of selecting between glucose and galactose and specificity for galactans arises in distinct branches of β-glucanases of the phylogenetic tree (Supplementary Fig. 5, the endo-β1,3-galactanases are another example of this), providing an example of convergent evolution. These finding suggest that the side activity seen in Amuc_0724 for some substrates with glucose in the −1 subsite is likely a relic of its evolutionary origin.

The second structural feature characteristic (but not unique) to O-glycanases is the requirement for accommodation of a β1,3-linkage between the −1 and −2 sugars. The structural features driving this specificity in the O-glycan active GH16 enzymes are identical to those in the GH16 enzymes specific for mixed linkage β-glucan[39]. An aromatic residue in the −2 subsite (also a part of the sequence motif from the subfamily) acts as a hydrophobic platform for the GlcNAc at this position and is at 90° relative to an aromatic residue carrying out the same function in the −1. For example, BACCAC_02680 residues W131 and W138 are platforms for the −1 and −2 sugars, respectively (Fig. 6b). This feature is conserved amongst β-glucanases (not in GH16 enzymes

with other activities) and is also required also for the degradation of polyLacNAc-based glycans.

Thirdly, in the O-glycanases, at the −2 subsite where the GlcNAc is accommodated, the N-acetyl group of the sugar faces away from the cleft and towards the solvent. Other non-mucinase GH16 enzymes with tighter clefts would not be able to accommodate this structure (for example see PDB 6JH5 and 4CTE). An additional observation about the −2 subsite and accommodation of sulphate groups resulted from the overlay of a porphyran product (originally from a porphyranase GH16 structure[40]) into the clefts of the O-glycanase GH16 enzymes. This indicates that sulfation on the GlcNAc at C3 could be accommodated within the cleft at the −2 subsite (Supplementary Fig. 20f).

The fourth structural feature likely driving O-glycan specificity concerns the +1 subsites. The substrate depletion assays support a preference for GlcNAc in the +1 subsite as the rate of degradation of TriLacNAc is much faster than that of milk oligosaccharides (Supplementary Fig. 10). Although the product complexes reported here do not have sugars in the positive subsites, comparison with previously published GH16-substrate complexes could be used explore how GlcNAc would sit in the +1 subsite and propose a structural rationale for the preference of GlcNAc at this position in the mucin active enzymes (Fig. 6d). BF4060 (and also its close homologue BACCAC_02680) has a significant preference for TriLacNAc over milk oligosaccharides (Supplementary Fig. 10) and analysis of the +1 subsite shows a narrow slot where a GlcNAc would insert with the N-acetyl pointing away from the cleft and S174 from Finger 5 would pincer the N-acetyl against finger 6, thus generating specificity for GlcNAc over glucose (Fig. 6d). The structures for BAC-CAC_03717 and Amuc_0724 are more accommodating of milk oligosaccharides and have more open +1 subsites.

Overall these data provide significant insight into the structural features that drive the polyLacNAc specificity of the GH16 O-glycan active enzymes characterised here.

**Generating O-glycan profiles from human tissue samples.** O-linked glycans are common modifications to proteins and lipids in addition to being the major component of mucins. Changes in O-glycosylation patterns in mucins and other glycoproteins have previously been detected in a variety of disease states and these changes can contribute to disease progression and severity, for example, by facilitating the metastasis of cancer. Therefore, these alterations represent promising biomarkers for the screening and prognosis of different diseases, especially in combination with other exo-acting enzymes of known specificity with activity against O-glycans. Indeed, sialyl Lewis X, sialyl lewis A and

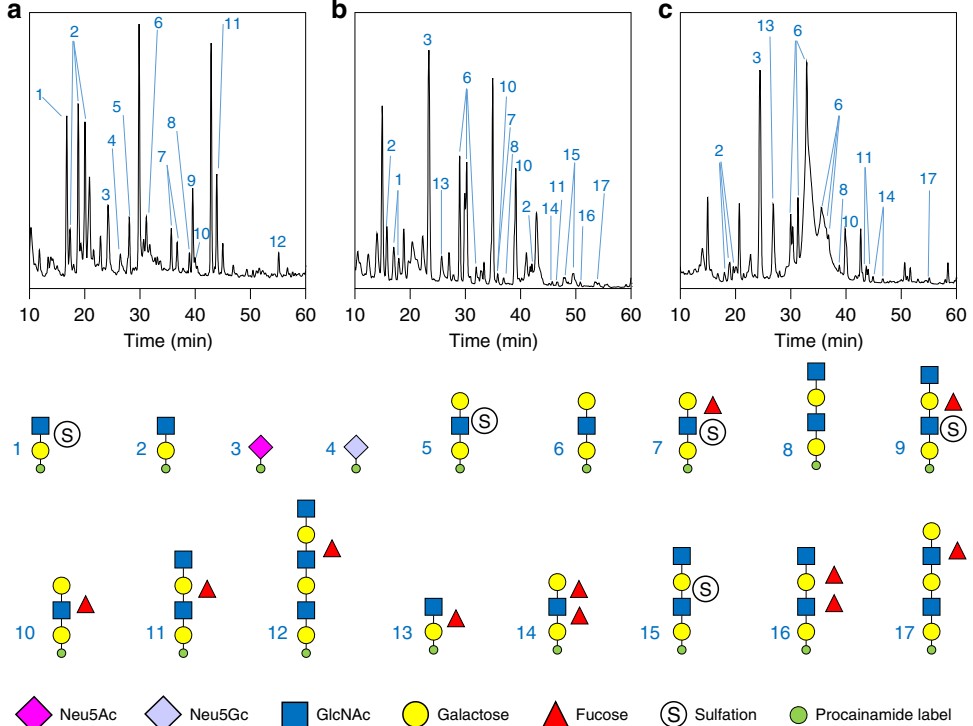

**Fig. 7 Examples of the O-glycan profiles that can be produced from a variety different human mucins by the Amuc_0724 O-glycanase.** Mucins from different samples were pre-treated with the broad acting sialidase BT0455[GH33] and then digested with the GH16 and products analysed by LC-FLD-ESI-MS. **a** Inflamed colonic tissue removed during a laproascopic panproctocolectomy from a patient with UC. **b** Bowel tissue from neonates with necrotising enterocolitis. **c** Colorectal cancer cell lines. Small amounts of Neu5Gc are seen in some of the UC samples (e.g., panel **a**) suggesting either the presence of contaminating dietary animal O-glycans remaining in the mucus layer or that this xenobiotic sugar has been incorporated into human mucins from dietary sources. The O-glycan profiles of all the different samples analysed are shown in Supplementary Fig. 21.

Forssman antigens are already used as biomarkers for different types of cancers[41–43].

To demonstrate that the GH16 enzymes could be used to generate O-glycan profiles for such screening purposes, we analysed the activity of one of the O-glycanases against mucins from three different types of human tissue/cells. Tissues from two adults suffering from ulcerative colitis (UC) were obtained as well as samples from preterm infants with necrotising enterocolitis (NEC); 4 infants of gestations 26, 27, 28 and 35 weeks. The mucus layer was scraped from the small amount of tissue (approximately 1 cm²) that could be spared by the pathologist and we used this to carry out assays. We also tested mucins produced by a number of cultured colorectal cancer (CRC) cell lines originating from different patients.

The human mucin samples were incubated with Amuc_0724, due to its broad activity, in combination with the sialidase BT0455[GH33] and the products were labelled with procainamide, and analysed using LC-FLD-ESI-MS. The data reveal that Amuc_0724 was able to release a range of Gal terminated and variably fucosylated and sulfated oligosaccharides from all of the samples tested, with similar structures to those identified from porcine SI mucin (Fig. 7 and Supplementary Fig. 21).

Whilst the presence of particular oligosaccharides in diseased vs healthy tissue was not assessed in these proof-of-principle experiments, the glycan profiles obtained demonstrate the potential of using these enzymes for either precision medicine or basic research applications using very small amounts of tissue. A total of 22 different oligosaccharides were detected in the various samples analysed and the glycan profiles observed varied significantly between samples (Fig. 7 and Supplementary Fig. 21). Furthermore, multiple peaks have the same sugar composition,

indicating the different peaks are due to variation in linkages between the sugars and that these can be differentiated using this approach (e.g. Fig. 7a, glycan 2). The variation in glycan profiles observed using the O-glycanase digestion suggest that these enzymes could be promising tools to discover disease-specific biomarkers.

## Discussion

Here we describe the characterisation of members of the GH16 family that have a specificity for the Galβ1,4GlcNAc linkage in the polyLacNAc chains found in mucins and other O-glycans. This discovery of endo-acting O-glycanases expressed by prominent mucin degrading gut bacteria is an important step forward in furthering our understanding of the complex relationship between the HGM and host.

We observed activity of these O-glycanases on both animal and human mucins from a range of tissues including the stomach and the small and large intestine. While much of the O-glycan that colonic microbiota will be exposed to will be from MUC2[44], as the major mucin expressed in the distal intestine, it is worth noting that these bacteria will also come into contact with significant amounts of MUC5AC, MUC5B and MUC6 mucins that have moved down the digestive tract from the saliva, oesophagus and stomach where they originated. In addition to these gel-forming mucins, gut microbes will be exposed to membrane-associated mucins that are a part of the apical surface glycocalyx of epithelial cells, especially when dead cells are sloughed off the epithelium throughout the GI tract and these include MUC3, MUC12 and MUC17[45,46]. Furthermore, greater than 80% of secreted proteins are O-glycosylated and the gut microbiota will come into contact with these from both host and dietary

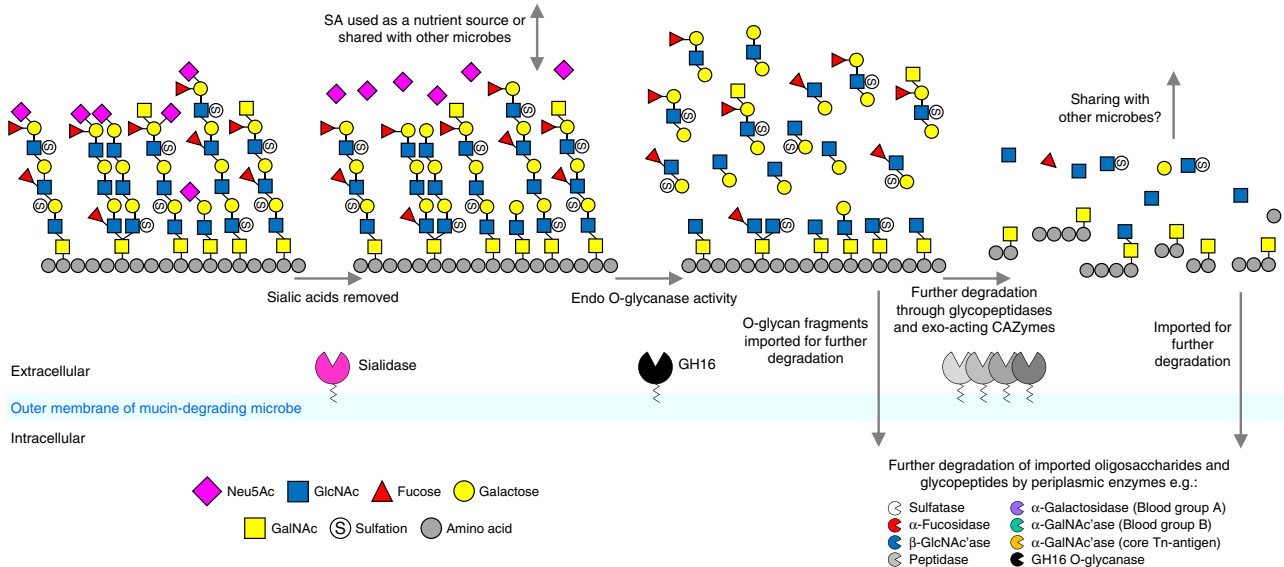

**Fig. 8 Model for the role of GH16 O-glycanases in mucin breakdown.** A model of the initial steps of mucin degradation on the surface of *Bacteroides spp.* and *A. muciniphila*. Sialic acid is removed by surface-localised sialidases and the GH16 enzymes remove oligosaccharides for import into the periplasm for further degradation by other CAZymes, including periplasmic GH16 O-glycanases. In *Bacteroides* species, glycan import is via energy dependent SusCD-like complexes, but in *A. muciniphila* the mechanism of glycan import across the outer membrane is unknown. The remaining mucin glycoprotein is likely further degraded by other extracellular CAZymes and glycopeptidases[19,29,50].

sources[47]. The activity of the GH16 enzymes against the broad range of substrates reported here indicate that these microbes can access O-glycans from the different types of mucins moving through the GI tract.

The discovery of this endo O-glycanase activity is in contrast to the previously proposed 'exo-trimming' model of mucin degradation and is more similar to pathways for degradation of other glycans seen in *Bacteroides* spp[18,48]. In all of the *Bacteroides* species studied here, at least one of the GH16 endo-mucinases is predicted to be a lipoprotein and whole cell assays support GH16 O-glycanases being localised to the cell surface. On the cell surface, the endo-mucinase and sialidase activities are likely the initial steps in O-glycan breakdown and other exo-acting CAZymes that trim capping sugars are also likely present (Fig. 8). The oligosaccharide products are then imported via outer membrane SusCD apparatus for further degradation in the periplasm. Several of the GH16 O-glycanases identified here are predicted to be periplasmic rather than cell surface. The role of these enzymes in the periplasm is likely to increase the efficiency of breakdown of the imported mucin oligosaccharides by increasing the number of chain ends available to periplasmic exo-acting glycosidases, a feature observed previously in glycan breakdown pathways[48].

Although targeting of polyLacNAc structures by the GH16 enzymes is likely an initial step in mucin breakdown, further processing of the remaining mucin would be required. The polyLacNAc side chains are attached to different core glycan structures, which are in turn linked to the peptide backbone. This is most likely dealt with through a combination of extracellular or surface exo-acting glycosidases and peptidases. Indeed, it has recently been shown that gut microbes, including *B. thetaiotaomicron*, express glyco-peptidases that specifically target polypeptides with O-glycan core structures still attached[49,50].

While this model applies to *Bacteroides* spp, it is currently unknown how *A. muciniphila* cells access complex glycans[6]. However, there is direct experimental evidence that at least one of the GH16 mucinases expressed by *A. muciniphila* (Amuc_2108) is localised to the outer membrane during growth on mucin, and

*A. muciniphila* cells also display O-glycanase-like activity on the cell surface, supporting a similar role for these enzymes in initiating mucin breakdown.

Structures of several O-glycanase enzymes revealed the characteristics of the subsites driving the specificity towards poly-LacNAc chains. In particular, alternating Gal and GlcNAc sugars, a requirement for a β1,3 linkage between the −2 and −1 subsites, and potential pockets for sulfation or fucosylation. There is a relatively low number of required subsites for catalysis, in contrast to that seen for GH16 enzymes with other activities, likely reflecting the highly heterogeneous nature of mucin O-glycans.

The combination of these new GH16 endo O-glycanase activities with other exo-acting CAZymes and the sensitive analytical techniques applied to human mucin samples described here demonstrates potential applications in both fundamental research and medicine. We hope this study facilitates partnerships between basic researchers and clinicians to explore the structures of O-glycans in a range of diseases on a larger scale in order to develop more effective biomarkers (e.g. for earlier detection of disease). This could lead to less invasive and more rapid techniques for diagnosis and monitoring remission in prevalent diseases like the ones explored here.

Overall, the findings reported here contribute significantly towards our understanding of the molecular mechanisms of mucin breakdown by the microbiota, a key process in maintaining the host-microbe symbiosis in the gut. These findings also open up the exciting possibility of exploiting this activity for characterisation and detection of biomarkers to allow more effective and earlier diagnosis of intestinal diseases such as IBD and CRC.

## Methods

**Sources of glycans and glycoproteins.** TriLacNAc was purchased from Elicityl and the rest of the defined oligosaccharides were from Carbosynth. PGM II and III (Sigma) was produced by dissolving in DI water at 50 mg ml$^{-1}$ and the precipitate removed by centrifugation before assays were carried out (leaving 35–40 mg ml$^{-1}$). Porcine small intestinal mucin was prepared as previously described with the only modification being a double CsCl gradient without Sepharose separation or SDS-PAGE in between[51]. Keratan was prepared as described previously[52].

**Bacterial strains**. The *Bacteroides* strains used were: *B. thetaiotaomicron* VPI-5482, *B. fragilis* NCTC9343, *B. caccae* ATCC43185, *B. cellulosilyticus* DSM14838, *B. finegoldii* DSM17565, *B. vulgatus* ATCC8483, *B. ovatus* ATCC8482, *B. intestinalis* DSM17393, and *Akkermansia muciniphila* ATCC BAA835/DSM22959.

**Cloning, expression and purification of recombinant proteins**. The DNA encoding the enzymes described in this report were amplified from genomic DNA and cloned into pET28b (Novagen) excluding the signal sequences, which were identified using SignalP 5.0[53]. Site-directed mutagenesis was carried out using Quikchange kit (Agilent). All recombinant enzymes were expressed in TUNER (Novagen) *E. coli* cells cultured in LB broth with kanamycin (50 µg ml$^{-1}$) at 37 °C in an orbital shaking incubator at 180 rpm. Cells were growth to mid-exponential phase (OD$_{600}$ ~0.6), cooled to 16 °C, 0.2 mM of isopropyl β-D-thiogalactopyranoside (IPTG) and incubated overnight at 16 °C in an orbital shaking incubator at 150 rpm. Cells were harvested, lysed by sonication, and the His-tagged protein was purified from the lysate by immobilised metal ion affinity chromatography using Talon resin (Clontech). The protein was bound to the resin, washed, and increasing amounts of imidazole used to elute the recombinant protein.

**Crystallisation**. For crystallography studies, the enzymes were purified using a further size exclusion chromatography step using a HiLoad Superdex 200 pg on an AKTA Pure FPLC system (GE Healthcare Life Sciences). SDS-PAGE gels were used to determine the pure fractions, which were then pooled and concentrated.

The GH16 enzymes were initially screened using commercial kits (Molecular Dimensions and Qiagen). Protein concentrations, crystallisation conditions and cryo-protectant used are given in Supplementary Table 6. The drops, composed of 0.1 µl of reservoir solution plus 0.1 or 0.2 µl of protein solution, were set up in sitting drop vapour diffusion plates by a Mosquito crystallisation robot (SPT Labtech) and incubated at 20 °C. BACCAC_02680$^{E143Q}$ was incubated with 5 mM of ligand for one hour and co-crystallised. BF4060 crystals were soaked in solution containing cryo-protectant and 3.5 mM TriLacNAc for 5 minutes prior to flash cooling in liquid nitrogen.

Data sets were integrated with XDS (Mar 2019)[54] or DIALS 1.14.5[55] or XIA2 0.5663[56] and scaled with Aimless 0.7.4[57]. Initial phases were obtained for Amuc_0724 by molecular replacement with Molrep 11.6.04[58] using 3WUT and Phaser 2.8.2[59] using a GH16 laminarinase from *Rhodothermus marinus* as search model (PDB 3ILN) for all the other proteins. Models were refined with refmac[60] and manual model building with Coot 0.8.9.1[61]. Final models were validated with MolProbity[62]. Other software used were from CCP4 suite[63] or Phenix suite 1.18.3855[64]. Figures were made with Pymol[65]. Validation of the sugar models was completed using Privateer MKIII[32]. The statistics from data processing and refinement can be found in Supplementary Table 5 and Privateer results can be found in Supplemental Table 7.

**Growth of bacterial species**. All growths were carried out in an anaerobic cabinet (Whitley A35 Workstation; Don Whitley). Glycerol stocks of bacteria were revived overnight in tryptone-yeast-extract-glycerol medium plus haematin[66]. *A. mucini-phila* and *B. xylanisolvans* required chopped meat broth (CMB) at this stage instead[11,24]. Monitoring growth against different substrates was done in minimal media for all *Bacteroides* spp.[8], however for *A. muciniphila* CMB was used without the addition of monosaccharides. For plate growths, a 96-well plate was monitored at 600 nm for 48 h by a Biotek Epoch plate reader. Growth against mono-saccharides and PGM II and III (precipitate removed) was carried out at 35 and 40 mg ml$^{-1}$, respectively. Growth against heparan sulphate and chondroitin sulphate was carried out at 20 mg ml$^{-1}$ and hyaluronic acid at 10 mg ml$^{-1}$ for viscosity reasons. GraphPad Prism was used to produce the figures.

**Recombinant enzyme assays**. For overnight assays, defined oligosaccharides were incubated at a final concentration of 1 mM in the presence of 3 µM of enzyme. For substrate depletion assays, 1 mM oligosaccharides were incubated with 0.1 µM enzyme and samples removed at various times. Some enzymes required increasing to 1 µM to assess the activity against substrates. The concentrations of different substrates are indicated to throughout the figures. All assays included 20 mM MOPS, pH 7, and were carried out at 37 °C. GraphPad Prism was used to produce the figures.

**Whole cell assays to determine cell surface activity**. 5 ml cultures (including 40 mg ml$^{-1}$ PGM type III) were grown on minimal media for *Bacteroides* species and modified CMB for *A. muciniphila*. Cells were harvested at mid-exponential, washed with PBS twice, and resuspended in 0.5 ml 2× PBS. Assays (200 µl) included 100 µl of cells, a final concentration of 1 mM Tri- or TetraLacNAc, and were incubated at 37 °C. Aliquots were removed over time and boiled to stop the reaction.

**Thin-layer chromatography**. For defined oligosaccharides and other polysaccharides, 3 µl of an assay containing 1 mM substrate was spotted on to silica plates (Merck; TLC Silica gel 60 F$_{254}$). For assays against mucin, this was increased

to 9 µl. The plates were resolved in running buffer containing butanol/acetic acid/water (2:1:1) and stained using a diphenylamine-aniline-phosphoric acid stain[67].

**Colorectal cancer cell line growth**. Human CRC cell lines were obtained from the Department of Surgery of the Leiden University Medical Center (LUMC), Leiden, The Netherlands. The cell lines cultured at the LUMC were kept in Hepes-buffered RPMI 1640 culture medium containing L-glutamine and supplemented with penicillin (5000 IU ml$^{-1}$), streptomycin (5 mg ml$^{-1}$), and 10% (v/v) foetal calf serum (FCS). Cells were incubated at 37 °C with 5% CO$_2$ in humidified air. The cells were harvested after reaching approximately 80% of confluence. To detach the cells from the culture flask a trypsin/EDTA solution in 1× PBS was used. Enzyme activity was stopped using the medium at a ratio 2:5 (trypsin:medium v/v). The cells were counted using TC20 automated cell counter from Bio-Rad technologies (California, USA) based on trypan blue staining. The cells were washed twice with 5 ml of 1× PBS, aliquoted to $2.0 \times 10^6$ cells ml$^{-1}$ of 1x PBS and pelleted by centrifuging 3 min at 1500g. Finally, the supernatant was removed, and the cell pellets were stored at −20 °C. Two million cells were used per reaction.

**Human sample collection**. IBD tissue samples were from two subjects. Matched ileal and colonic samples were obtained from one panproctocolectomy and one ileocaecal resection. Samples were transferred on wet ice directly to the laboratory for mechanical isolation of the mucus layer by gently scraping using a pipette tip. For NEC samples, fresh tissue was collected from surgically resected specimens when a clinically necessary procedure was taking part, stored briefly in sterile phosphate buffered saline and transported to the laboratory on ice.

**HPAEC-PAD**. To analyse the substrate depletion assays, sugars were separated using a CARBOPAC PA-100 anion exchange column with a CARBOPAC PA-100 guard. Flow was 1 ml min$^{-1}$ and elution conditions were 0–10 min, 100 mM NaOH; 10–35 min 100 mM NaOH with a 0–166 mM sodium acetate gradient. The software was Chromeleon Chromatography Data System.

**Procainamide labelling**. Reducing ends of GH16 products were labelled by reductive amination using a procainamide labelling kit containing sodium cyanoborohydride as reductant (Ludger). Before and after labelling the O-glycan samples were cleaned up using PBM plates and S-cartridges, respectively (Ludger).

**LC-FLD-ESI-MS of procainamide labelled glycans**. Procainamide-labelled samples were analysed by LC-FLD-ESI-MS. 25 µl of sample was injected to a Waters ACQUITY UPLC Glycan BEH Amide column (2.1 × 150 mm, 1.7 µm particle size, 130 Å pore size) at 40 °C on a Dionex Ultimate 3000 UHPLC instrument with a fluorescent detector ($\lambda_{ex} = 310$ nm, $\lambda_{em} = 370$ nm) attached to a Bruker Amazon Speed ETD. Mobile phase A was a 50 mM ammonium formate solution (pH 4.4) and mobile phase B was neat acetonitrile. Analyte separation was accomplished by gradients running at a flow rate of 0.4 ml min$^{-1}$ from 85 to 57% mobile phase B over 115 min and from 85 to 62% over 95 min for mucin and keratan samples, respectively. The Amazon speed was operated in the positive sensitivity mode using the following settings: source temperature, 180 °C; gas flow. 41 min$^{-1}$; capillary voltage, 4500 V; ICC target, 200,000; maximum accumulation time, 50.00 ms; rolling average, 2; number of precursor ions selected, 3; scan mode, enhanced resolution; mass range scanned, 400 to 1700. HyStar v3.2 was used for data collection of chromatography and mass spectrometry. GraphPad Prism was used to produce the figures.

**Analysis of mass spectrometry data**. Mass spectrometry of procainamide-labelled glycans was analysed using Bruker Compass Data Analysis Software and GlycoWorkbench[68]. Glycan compositions were elucidated on the basis of MS$^2$ fragmentation and previously published data.

**Bioinformatics**. Putative signal sequences were identified using SignalP 5.0[53]. Sequence identities were determined using Clustal Omega using full sequences[69]. The IMG database (https://img.jgi.doe.gov/) was used to analyse synteny between different species[70]. The CAZy database (www.cazy.org) was used as the main reference for CAZymes[71]. To determine the boundaries between different modules in a protein Pfam[72] and SMART[73,74] were used.

Alignments and phylogenetic trees were completed in SeaView[75]. Sequences were aligned in SeaView using Clustal. Gblocks were applied to: allow smaller final blocks, allow gap positions within the final blocks, and allow less strict flanking positions. Trees were built using PhyML with aLRT (SH-like) branch support, model-given amino acid equilibrium frequencies, no invariable sites, optimised across site rate variation and the best of NNI and SPR tree searching operations.

**Reporting summary**. Further information on experimental design is available in the Nature Research Reporting Summary linked to this paper.

## Data availability

The crystal structures are deposited in the Protein Data Bank under the accession numbers 6T2N, 6T2O, 6T2P, 6T2Q, 6T2R and 6T2S. The other data that supports the findings in this paper are available in the Source Data file or upon request from the corresponding authors.

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

## Acknowledgements

We thank Carl Morland (Newcastle University, UK) for his expert technical assistance. We thank Dr. Mirjam Czjzek for her expert advice on the structural data and Prof Robert Hirt for his insightful conversations about phylogenetics. We would like to thank Diamond Light Source (Oxfordshire, UK) for beamtime (proposal mx18598) and staff of beamline I03, I04-1 and I24. We are grateful to Newcastle Biobank and NIHR Newcastle Biomedical Research Centre. Dr. Jose Muñoz-Muñoz kingly gifted the arabinogalactan substrates used. The NEC samples were collected as part of the ethically approved SERVIS study (REC 10/H0908/39). The colorectal cancer cell lines were from the Department of Surgery, Leiden University Medical Centre (LUMC), Leiden. The work was funded by a BBSRC/Innovate UK IB catalyst award 'Glycoenzymes for Bioindustries' (BB/M029018/1).

## Author contributions

L.I.C. sorted through previously published gene upregulation and protein expression data. L.I.C. and M.V.L. carried out reactions on defined oligosaccharides and commercial polysaccharides available. M.V.L. produced catalytic mutants. L.I.C. and M.V.L. purified proteins for crystallography. M.V.L. and A.B. obtained and harvested crystals, collected data, and solved crystal structures. L.I.C. and M.V.L. carried out comparisons of crystal structures with ones already present in the database. P.A.U. carried out the LC-MS. L.I.C. analysed the LC-MS data. D.I.R.S. provided resources. P.C. and J.P.P. collected and purified porcine small intestinal mucin. F.Z. and R.J.L. purified keratan sulphate. R.G. and E.C.M. supplied the *B. thetaiotaomicron* PUL knockout strains. L.I.C. carried out substrate depletion assays, carried out growth experiments with different bacterial species, completed the phylogenetic analysis, carried out assays against human samples, and assays of with other enzymes used in the report against defined oligosaccharides. C.A.L., R.R.B., M.D., and S.N. were responsible for ethical approval, governance, patient identification, and sample collection for IBD tissues. R.R.B. performed surgery where adult intestinal samples were collected. K.C. and C.A.L. were responsible for lab preparation of IBD tissue. C.J.S. and J.E.B. were responsible for ethical approval and provision of samples for the preterm neonate NEC samples. J.E.B. performed the neonate surgery. K.M. prepared the colorectal cancer cells and M.W. provided resources. L.I.C. and D.N.B. designed the experiments, analysed the data, and wrote the manuscript.

## Competing interests

The authors declare no competing interests.

## Ethics declaration

Adult IBD tissue was collected as part of the Newcastle Biobank following written consent according to approval from Newcastle and North Tyneside Research Ethics Committee 1 (REC:17/NE/0361). Necrotising enterocolitis samples were collected with written consent obtained from parents and ethical permission provided through the SERVIS study (approvals 10/H0908/39 and 15-NE-0334).
