## [Peer Review File · Nature Communications]

Reviewers' comments:

Reviewer #1 (Remarks to the Author):

The colonic mucus barrier separates the human gut microbiome (both commensals and pathogens) from the epithelium and plays a vital role as primary defense against the enteric pathogens. Recent literature underscores the importance of this barrier for the host immune system and thus, the crosstalk between the microbiome and the mucus barrier is a decisive event in health and disease. The microbiome is armored with an array of enzymes that specifically cleave variety of linkages in the host mucus. So far, this model of microbial metabolism of host-secreted mucus involves only exo-acting enzymes, meaning that the enzymes are known to act on the linkages connecting the terminal sugars of the mucin glycoprotein.

Here, Crouch et al. report discovery of endo-acting O-glycanases in select human gut bacterial species belonging to two major phyla Bacteroidetes (3 species; *Bacteroidetes thetaiotaomicron*, *B. fragilis* and *B. caccae*) and Verrucomicrobia (*A. muciniphila*). First, the authors use data from previously published studies from two other labs that reported higher expression of certain transcripts/proteins belonging to the glycoside hydrolase (GH) family 16. Next, they used mass spectrometry to demonstrate that cloned protein products of these transcripts act on mucus polysaccharides (porcine gastric mucus) in an endo-acting manner. The authors then performed a detailed biochemical characterization (including analysis of the crystal structures) of these proteins to further support the endo-acting nature of these enzymes on the colonic mucus. Finally, using limited samples from human patients (ulcerative colitis, necrotising enterocolitis) and human-derived colorectal cancer cell lines, the authors show that the endo-acting ability of these enzymes also extends to human-derived mucins.

The study is certainly informative and the discovery of endo-acting glycanases sheds important light on the mechanisms behind how the gut microbiome interacts with the mucus barrier. One major issue is that the patient sample size is small. Given this, the patient data cannot be conclusive – I suggest that either the authors increase their cohort size or exclude the patient data from the current manuscript or do not make definitive conclusions from the patient data. To improve quality of the manuscript, I have several comments to re-formulate the text.

General comments

Identification of GH16 enzymes expressed during growth on mucin: The authors have used data from three previous studies (Martens, *Cell Host Microbe*, 2008; Desai, *Cell*, 2016; and Ottmann, *Applied and Environmental Microbiology*, 2017). It might be useful to explain the rationale behind shortlisting some specific transcripts and not others and also why only these four specific species were chosen. Fig. S21 shows that there are also other Bacteroidetes strains that can grow on mucus glycans. Although Fig. 1 is useful to understand how bacteria metabolize the mucus polysaccharides, as this figure does not have new data, it might be good to have it as a supplemental figure. Authors can use previously published reviews (e.g., Martens, *Nature Reviews Microbiology*, 2018) to highlight these biochemical aspects related to mucus degradation. Instead, it might be useful to provide some details of the localization of the identified GH16 enzymes within the bacterial cell wall. This should be generally explored in more detail to offer or characterize a mechanism. How mucin O-glycans are degraded by the endo-hydrolyases (first, OM-attached enzyme X cleaves sugar residue Y, then transport into periplasmic space, then enzyme Z catalyzes; something like this). Amuc_2108 seems to be located in the OM, while other GH16 enzymes (Amuc_0724 and Amuc_0875 seem to be either in the periplasmic space or in the IM?).

GH16 enzymes are endo-acting mucinases:

Fig. 2a: Generation of recombinant forms of the 9 identified GH16 enzymes from BT, BF, BC and AM. They test their activity to hydrolyze either porcine SI mucin, porcine gastric mucin type PGM II and porcine gastric mucin type PGM III.

Comment: Do the recombinant forms of the enzymes still contain their lipid anchors? They refer to Fig. S8. What do they show there? Is this a thin layer chromatography?? What is the control?

Figure 2b: Then, they chose the products of BF4058 of SI mucin for further characterization.

Comment: why this enzyme and why SI mucin?

Products were treated with an exo-acting hydrolyse.

Comment: It's unclear why they do this experiment and what is the conclusion. Everything they "conclude" is "likely" or "possibly".

Furthermore, they have already investigated the structure via MS/MS...

Fig. 2B: Please clarify what is meant by the red star?

Next, they tested the capability of the 9 enzymes to degrade keratan sulfate, which is structurally similar to mucin O-glycans. They detect that the 9 enzymes are active against keratan sulfates and that the enzymes can tolerate "a significant number of sulfates".

Comment: Why is this experiment done? What about the sulfates in the mucins they used in previous experiments?

Next, they used a range of defined oligosaccharides to assess the substrate specificity of the 9 enzymes

Comment: Data and text are sound. However, a combining or concluding main figure would be nice for the main manuscript

Next, they tested the hydrolysis activity of the 9 enzymes towards previously published validated GH16 substrates

Comment: They could not really detect any hydrolysis of these substrates (besides some exceptions, which are discussed in the SM)

But: what is the conclusion? GH16 enzymes with specificity for mucins? What's the difference and the final conclusion?

Overall, this paragraph is not really clear and comprehensive. The data of Fig. 2A are nice. However, it should be made clear what the purpose of Fig. 2B is. Why this is done and what are the conclusions of the "story". The text should mention that the red asterisks refer to the disappeared peaks.

Crystal structures:

Comment: Perhaps here they should clearly mention that this is done to elucidate the structural basis for mucin-specificity?

They analyzed the apo-structures of BC2680, BC2680E143Q, BC3717 and AM0724.

Comment: Why were they chosen? Why the mutant?

Additionally, BF4060 and BC2680E143Q together with a trisaccharide.

Comment: What is the purpose of this? Why now adding BF4060? Why not comparing BC2680 with its mutant? Is this a catalytic mutant? Why not only perform co-crystallization of the holo-enzyme together with the substrate? Where does the trisaccharide come from? Was it added to the enzyme in its trisaccharide form or is it the product of an enzymatic reaction by these enzymes?

Activity against human mucins:

Comment: It should be mentioned in the text how the "human mucin samples" were generated and processed.

They incubated the human mucin samples with AM0724 (why only this one?) and sialidase. Products were then labeled and identified.

They conclude that "enzymes could be used to characterise the glycan structures present in

mucins from different disease states”

Comment: This conclusion is an overinterpretation and not supported by the available data. Is there a real difference between different disease states? There are no samples from healthy controls and only 2 samples from UC patients. The sample choice (uninflamed ileum and inflamed colon from Patient 1; inflamed ileum, uninflamed ileum and uninflamed colon) seems to be very random (and n=1 for each condition). The overall conclusion “showed different patterns of oligosaccharides released depending on the disease and tissue type” is not supported by the data. There are too many variables (individual, disease type, inflammation state, tissue sample, ...) and such a conclusion cannot be made by analyzing samples from only 1-2 individuals with a random sample selection. I am not certain about showing these data with definitive conclusions. I suggest to beef up the cohort size or tone down the conclusion based on the human data (including data on the CRC cell lines).

Reviewer #2 (Remarks to the Author):

This manuscript by Crouch et al. on the ability of O-glycanases from human gut microbiota members to digest mucin O-glycans via an endo-acting mechanism is well-supported by analytical and X-ray crystallographic studies. Some additional experiments and discussion are suggested to enhance the impact of this manuscript.

Major claims: (1) Gut bacteria access mucin O-glycans in the human intestine as a nutrient source. (2) Instead of acting as exoglycosidases, the enzymes secreted by these gut bacteria act via an endoglycosidase mechanism. (3) Understanding this mechanism is key to understanding human gut microbiota interactions. (4) Recombinant O-glycanases can be useful tools to explore changes in the O-glycome of IBD and colorectal cancer.

Evaluation: There is significant interest in the human gut microbiome, and this manuscript describes detailed biochemical and structural evidence that glycosidases of the GH16 family from *Bacteroides* and *Akkermansia* act by digesting mucin glycans from porcine intestine via an endo mechanism instead of exo. Biochemical evidence in the form of glycan LCMS is adequate, although the authors are suggested to further simplify Figure 2 for the non-glycoscientist. In particular, it is unclear why it is significant to group the enzymes based on products of more or less than four sugars (seems arbitrary). It would also be helpful to compare the structures of human versus porcine intestinal mucin to provide relevance to the study, as well as gain a biochemical understanding of the expression levels of the enzymes described in the bacteria specified (only previously reported transcriptomic dataset is referenced). The cloning and expression of the recombinant proteins should also be described (right now the reader is referred to reference 25). Enzymes used in this study were produced in an *E. coli* mutant, but how (if any at all) abundantly are these enzymes expressed, and how abundant are the requisite sialidases? Are these enzymes also active on N-glycans with extended poly-LacNAc chains? N-glycans are also prominent members of the gut glycome. Is there specificity at the protein level as to which mucin glycoproteins are degraded by these enzymes (doesn't seem to, but explicit mention of this point would be helpful). Can the SusC/D outer membrane complex be genetically manipulated to show that there is indeed uptake of the broken down glycan byproducts that could then be used for nutrition? More sophisticated experiments that more closely mimic the nature of the human gut microbiome would greatly enhance this manuscript. Is mucosal viscosity a factor in the activity? What about pH? The pH in the small intestine is ~ pH 6 and ~7.4 in the terminal ileum.

Overall, these findings are of specific interest to the glycoscience-gut microbiome community, and the conclusions are original, but would be further strengthened by the suggested experiments. There is additional impact in adding information to carbohydrate-active enzymes. While the authors' experimental conclusions are well supported, subjectively, I find a large disconnect in this small detail (exo vs endo mechanism) and it being incredibly significant and key to understanding

gut-microbe interactions. It is not clear that this is a highly debated topic that is of general widespread interest to the wider community and that it is a key step to influence thinking in the field (although a better discussion of these specific topics could influence my evaluation). In my opinion, there is greater impact on the utility of these enzymes as tools to determine changes in patient samples, but there is not adequate data here for comparison (healthy samples are limited). Again, figure 4 can be better represented as digested data with bar graphs and statistical analysis instead of a series of chromatograms.

Statistical analysis: Although the chromatograms used for glycan analysis are included and the description of methods used to generate them are satisfactory, there are limited mentions of any statistical analysis used. Instead of the chromatograms, bar graphs depicting the major conclusions replete with statistical measures of reproducibility are suggested.

Reviewer #3 (Remarks to the Author):

This manuscript that reports a substantial volume of work on the identification and detailed characterization of the first endo-acting mucinases from a number of species of prominent human gut bacteria spanning two distinct phyla. Mucin samples from human were also shown to be degraded by one of the characterized enzymes. These endo-mucinases represent the first reported of such specificity in the large and well-studied glycoside hydrolase family 16, making this work of great interest to the carbohydrate enzyme community as well as to the gut microbiota community.

Regrettably, the presentation of the manuscript leaves much to be desired, and in many instances a clear lack of attention to detail is evident. It would seem that that the numerous co-authors have not carefully proofread the manuscript, which is burdensome to reviewers and regrettable considering the quality of the journal the authors are targeting. It is incumbent upon the corresponding authors to ensure that this is done properly.

The following comments must be addressed. These are presented generally in order of appearance in the documents.

Main text

Typographical/grammatical/sentence-structure problems must be addressed, for example on lines: 57, 58, 106, 116, 182, 278, 321, 354, 372, 378, 441, 445, 465, 499, 520. Note that this does not comprise an exhaustive list and the manuscript must be checked carefully throughout.

The sections titled "The GH16 enzymes are endo-acting mucinases" and "Crystal structures of O-glycan active GH16 family members" are big sections that present the bulk of the data; their content should be divided into smaller subsections with informative headers to improve readability.

It would be more logical for the section titled "Phylogenetic analysis of the GH16 O-glycanases" to be moved up, directly after the section "Identification of GH16 enzymes expressed during growth on mucin". Also, this way it is consistent with the figure order and supplementary discussion order. Content that is best presented after the enzymology and structural biology sections can go in the discussion.

For the section titled "The GH16 O-glycanases are active against human mucins from healthy and diseased samples", more interpretation of the results is required. The first paragraph describes the sample source, the second paragraph is discussion material (should really be moved to the discussion section), and only the third paragraph (lines 327-333) describes the results. Maybe elaborate a bit on the pattern of oligosaccharides released as a function of disease state. In light of the possibility of use as a biomarker, are there perhaps any particular unique or uniquely enriched oligosaccharides corresponding to a disease?

Line 95: Define "Sus" and explain.

Line 109-112: with regard to ref. 29, it would seem that the linkage specificity of the enzymes in the present study is not entirely unprecedented/surprising. Here, and in the corresponding structural biology sections, the structural similarities and differences with regard to carbohydrate recognition should be explicitly described, including through the use of structural superpositions.

Line 114: What does "sequence diverse" mean in this context? Presumably, a subfamily would have comparatively low diversity compared with the larger GH16 family.

Line 128: PGM = porcine gastric mucin.

Line 144: Please elaborate on the structural basis for this specificity, especially with regard to the cleft-like active-site of these enzymes (see also comment on Lines 109-112 above).

Line 154: Where is the data that shows this? I think it is supposed to be in Supplemental Figure 13 but is missing (see below).

Line 155: Was there a particular reason for the selection of BT4058 for detailed product analysis?

Line 212-217: There are six structures described here but only stats for four of the structures in sup table 5.

Line 217: Sup figs 17-19 not 15-17.

Line 214-218: Privateer [Nature Structural & Molecular Biology volume 22, pages 833–834 (2015)] must be used to validate these sugar models and the resulting output reported in a supplemental table. (See also identical comment in SI section below).

Line 276: Meaning unclear (misplaced modifier).

Line 271: Sup fig 16 not 18.

Line 281: Run-on sentence/poor grammar.

Line 327: Was there a particular reason for the selection of Amuc_0724 for assays with human mucin samples?

Line 435: Which is it, soak or co-crystallized? If both were tried, which led to the structure presented?

Line 456-457 and rest of methods section: the preferred abbreviation for milliliter is mL not ml (same for microliter; uL not ul)

Figure 2: It should be clearly stated what the control is. Just sialidase?

Figure 3: Baccac_02680 structure should be labelled as the E143Q mutant. In panel b, why are the catalytic residues colored red in Bf4060 but not in Baccac_02680? Be consistent. Similar issue in sup fig 19. Check caption carefully – Beta1,3-linkage, "insert number here", the supplementary figure you want to refer readers to is 19 not 17.

Supplemental Information

Typographical/grammatical/sentence-structure problems must be addressed on lines: 110, 166, 226, 265, 301, 504

Line 190-198: Please cut down on these texts which are redundant with lines 193-205 from the main text. The supplemental discussion should complement the main text instead of rephrasing the same thing.

Line 209-213: These texts are likewise redundant with main text lines 181-192.

Line 30-527: Carefully go through every section and remove all instances of redundancy with the main text.

Line 316-323: I'm curious how convincing the electron density that supports skew boat conformation is at 2.7 Å resolution for Baccac_02680. There is not a single figure with electron density maps; 2Fo-Fc maps at a minimum, and ideally composite omit maps, should be incorporated into figure.

Line 316-323: Privateer [Nature Structural & Molecular Biology volume 22, pages 833–834 (2015)] must be used to validate these sugar models and the resulting output reported in a supplemental table.

Line 321-323: So the density was too poor to define the ligand in BF4060 and it was built anyway? Same comments as previous: please show appropriate maps and run Privateer validation.

Line 345-346: What does "different evolutionary origins" mean here? Presumably all enzymes in this subfamily have a common ancestor (hence the ability to construct a meaningful phylogeny)?

Table 3: The "prefixes used for brevity" are really superfluous and should be replaced with the "true locus tag prefixes" throughout. In most cases they are the same, or do not result in the remove of many characters. Is this just laziness regarding figure editing?

Table 5: Completely missing all statistics for apo Baccac_02680 and Baccac_02680 E143Q. For Amuc_0724 and BF4060 were there really zero water molecules? Missing Ramachandran stats for BF4060. Table caption: "reflections that were excluded from.....".

Table 6: What is L404 ligand?

Figure 4: Please make the distinction between e) and f) clearer in the figure caption because they are both mixed-linkage glucan. Xyloglucan in i) is inaccurate.

Figure 5: Label figure panels

Figure 6 and 7: I am unable to find a look-up table as described in the legend. Regardless, most of the branch labels are impossible to read. Overall, these trees should be presented with an alternate layout that avoids overlap; a rectangular format would be far preferred, using collapsed branches where advantageous. A properly organised and displayed tree would not require all of the accessory text annotations and arrows.

Figure 6 and 7: Insufficient information is provided regarding the algorithms used for sequence alignment and phylogeny estimation. Presently, there is no information on the statistical significance of any of the "Branches" described. At a minimum, a Maximum Likelihood phylogenies should be presented, using at least 100 bootstraps, and significant bootstrap values must be indicated on the published trees. A suitable outgroup should be used to root the trees; in the case of Figure 7, presumably one or more related GH16 subfamilies could be used.

Figure 6 and 7: Please provide the input multiple sequence alignment files in FASTA format and trees in a suitable format (e.g. Newick, PhyloXML) as supplemental information files to allow

reader direct access to the underlying data.

Figure 8: The last sentence of the caption is slightly confusing. Please make it clear that BT0455 is the sialidase and GH95 is the fucosidase.

Figure 11: Is species 14 correctly depicted? There is a stray GlcNAc on the end of a row of sulfations.

Figure 13: Panel E is missing and the panel labels on the figure are not consistent with their description in the figure caption. Is low-activity Amuc_0875 the missing data here?

Figure 15: The lanes are not labeled.

Figure 16: μM enzyme not μm , it is a unit of concentration not length.

Figure 16: Negative results determined only by TLC analysis is not particularly compelling, especially given weak staining efficiency (high limit of detection) and some streaking at Rf values consistent with oligosaccharides is observed. Supporting data using a standard reducing sugar activity assay must be provided to quantify relative specific activities.

Figure 17: It would be helpful if the relevant loops are labeled with their finger number. Also, consider moving this figure to after 18 and 19 as its content is first referred to from the text after first reference to 18 and 19.

End comments

Dear Reviewers,

Thank you for your time and thoughtful insights.

Based on the comments we have made substantial changes to the MS as detailed in the bullet point list below and the response to individual reviewer's comments.

One of the main issues was the way we had presented the data from different types of human tissues led to the significance being lost. The aim of those small experiments was to demonstrate how the GH16 enzymes may be useful in facilitating more extensive and larger scale studies into the O-glycans structures present in different disease states. This point has been elaborated on in the responses to reviewer's comments and the text modified.

The major changes we have made to the manuscript include:

- Re-ordering the Results section and breaking it into small parts.
- Included new data showing endo activity on the cell surface Fig 5 and a section into the main text.
- Deleted most of Fig 1
- Added in the genomic context of the upregulated genes and the enzyme modules
- Moved Fig 2b to supplementary and the corresponding text has been simplified and moved to supp discussion
- Fig 2a is now Fig 3
- Fig 4 (now Fig 7) has also been slimmed down and the full data put into the Supplementary
- A heat map has now been included to demonstrate the specificities of the enzymes (now Fig 4)
- Included omit maps of the ligands for the complexed structures and completed a Privateer validation of them (Supp Fig 17)
- The phylogenetic trees have been re-done, details of how they were produced were elaborated on, and the text files for the sequences used has been included

We hope you think that the changes solve all the issues raised and the manuscript now flows better for the reader.

Best wishes,
Lucy and Dave

Reviewer #1 (Remarks to the Author):

The colonic mucus barrier separates the human gut microbiome (both commensals and pathogens) from the epithelium and plays a vital role as primary defense against the enteric pathogens. Recent literature underscores the importance of this barrier for the host immune system and thus, the crosstalk between the microbiome and the mucus barrier is a decisive event in health and disease. The microbiome is armored with an array of enzymes that specifically cleave variety of linkages in the host mucus. So far, this model of microbial metabolism of host-secreted mucus involves only exo-acting enzymes, meaning that the enzymes are known to act on the linkages connecting the terminal sugars of the mucin glycoprotein.

Here, Crouch et al. report discovery of endo-acting O-glycanases in select human gut bacterial species belonging to two major phyla Bacteroidetes (3 species; Bacteroidetes thetaiotaomicron, B. fragilis and B. caccae) and Verrucomicrobia (A. muciniphila). First, the authors use data from previously published studies from two other labs that reported higher expression of certain transcripts/proteins belonging to the glycoside hydrolase (GH) family 16. Next, they used mass spectrometry to demonstrate that cloned protein products of these transcripts act on mucus polysaccharides (porcine gastric mucus) in an endo-acting manner. The authors then performed a detailed biochemical characterization (including analyse of the crystal structures) of these proteins to further support the endo-acting nature of these enzymes on the colonic mucus. Finally, using limited samples from human patients (ulcerative colitis, necrotising enterocolitis) and human-derived colorectal cancer cell lines, the authors show that the endo-acting ability of these enzymes also extends to human-derived mucins.

The study is certainly informative and the discovery of endo-acting glycanases sheds important light on the mechanisms behind how the gut microbiome interacts with the mucus barrier. One major issue is that the patient sample size is small. Given this, the patient data cannot be conclusive – I suggest that either the authors increase their cohort size or exclude the patient data from the current manuscript or do not make definitive conclusions from the patient data.

We thank the reviewer for their supportive comments about our work.

The purpose of testing the different human samples was to demonstrate that it is possible to analyse the O-glycans from small amounts of tissue and cells. There was no attempt to draw conclusions about diseases. Hopefully, these enzymes will potentially be useful to people who work on disease states where changes in O-glycosylation occur. These small pieces of tissue are the size that can readily be obtained from surgery or biopsies, for example, so the data we present is to show that this combination of techniques and analysis could be used by other researchers in partnership with biobanks and clinicians to study a number of diseases associated with the GI tract. The paperwork and logistics to obtain these samples was quite substantial as it was new, so before reaching towards more ambitious sampling projects we wanted to demonstrate whether the approach was viable. Hopefully publications like this can be used to encourage support for the development of biobanks to allow bigger cohorts to be analysed.

We agree the sample sizes would need to be much larger for useful diagnostic data to be obtained from the patterns of healthy vs diseased O-glycans. It would also be very expensive and time consuming to screen enough samples to provide enough meaningful data for statistics to be applied, so this would have to be a separate project. For these reasons we feel the data as it is worthy of keeping in the paper and our intentions are clarified better in the text now.

To improve quality of the manuscript, I have several comments to re-formulate the text.

General comments

Identification of GH16 enzymes expressed during growth on mucin: The authors have used data from three previous studies (Martens, *Cell Host Microbe*, 2008; Desai, *Cell*, 2016; and Ottmann, *Applied and Environmental Microbiology*, 2017). It might be useful to explain the rationale behind shortlisting some specific transcripts and not others and also why only these four specific species were chosen.

The four species were chosen as they have all previously been shown to be prominent mucin degrading members of the normal gut microbiota and are as far as we are aware the main Gram negative members of the microbiota that have had their transcriptomes and/or proteomes analysed during growth on mucins. To our knowledge these are all the 'omics datasets available for growth of these species on mucins. Certain Firmicutes and Actinobacteria (Bifidobacteria) can also grow on mucins but analysis of the genomes of these spp showed they do not encode Gh16s.

In terms of choosing the transcripts, for the *Bacteroides* spp. we included transcripts that had a 10-fold increase or above as a starting point, which is where the previous studies applied their cut-off for 'upregulation'. From there we expanded the transcript pool to include the whole predicted PUL, so if some genes in a PUL were highly upregulated but the others were not they were still included in the figure. Furthermore, for the *B. thetaiotaomicron* data, if genes were upregulated in one growth phase and not the other, they were included in both graphs.

For *A. muciniphila*, we used a starting point of 5-fold for the in vitro transcriptomic data sets. The fold increase in upregulation was low relative to what was observed for *Bacteroides* spp., which may be due to mucin being the favoured substrate in this species and a high basal level of expression. The Supplementary Info has been modified to make all these points clearer.

Fig. S21 shows that there are also other Bacteroidetes strains that can grow on mucus glycans. We included other species in the growth studies to provide a better idea about how widespread this trait is and how well different species grow on mucin. Growth data can then be compared with other observations, such as the gene content of different species, for example. The Supplementary Info has been modified to make all these points clearer.

Although Fig. 1 is useful to understand how bacteria metabolize the mucus polysaccharides, as this figure does not have new data, it might be good to have it as a supplemental figure. Authors can use previously published reviews (e.g., Martens, *Nature Reviews Microbiology*, 2018) to highlight these biochemical aspects related to mucus degradation.

The intention of Fig.1 was to provide a non-specialist with context and for clarity for all readers. We agree that there are some nice reviews out there covering similar details presented in 1a and b, but we still feel it is important that readers have this information available to them here for clarity and to aid in understanding. After consideration of the reviewers point though we have removed Fig 1a and kept 1b. Fig 1 panel c has been transformed into Fig. 8, which we have tried to make more detailed and informative and provide a summary of our main findings. Hopefully the reviewer agrees that these changes help with understanding of the work and are worth keeping.

Instead, it might be useful to provide some details of the localization of the identified GH16 enzymes within the bacterial cell wall. This should be generally explored in more detail to offer or characterize a mechanism. How mucin O-glycans are degraded by the endo-hydrolyases (first, OM-attached enzyme X cleaves sugar residue Y, then transport into periplasmic space, then enzyme Z catalyzes; something like this). Amuc_2108 seems to be located in the OM, while other GH16 enzymes (Amuc_0724 and Amuc_0875 seem to be either in the periplasmic space or in the IM?).

Figure 1 has mostly been deleted and your suggestions were used to make Figure 4 and 8. Here we have tried to show where the GH16 enzymes are localised as well as their role in the mucin degradation pathway based on what we know about other mucin-active enzymes. Hopefully this makes the bigger picture clearer for the reader and placing this at the end of the MS provides a more useful summary of the main findings.

The localisation GH16 enzymes are endo-acting mucinases:

Fig. 2a: Generation of recombinant forms of the 9 identified GH16 enzymes from BT, BF, BC and AM. They test their activity to hydrolyze either porcine SI mucin, porcine gastric mucin type PGM II and porcine gastric mucin type PGM III.

Comment: Do the recombinant forms of the enzymes still contain their lipid anchors?

No. As a standard practise we clone enzymes without their Type II signal sequence so they are expressed in E .coli in the cytoplasm. Details have been added to the methods.

They refer to Fig. S8. What do they show there? Is this a thin layer chromatography?? What is the control?

We state that the Supp Fig. 8 is thin layer chromatography (TLC) in the main text. The TLC shows products being released relative to the two controls. The two controls are 1. No enzymes 2. With sialidase and fucosidases alone. This information is in the legend.

Figure 2b: Then, they chose the products of BF4058 of SI mucin for further characterization.

Comment: why this enzyme and why SI mucin?

We chose the small intestinal mucin as it is more highly purified than the commercially available PGM and thus the products can be more accurately characterised. BF4058 was chosen as this is one of the enzymes predicted to be cell surface located and thus more likely to be involved in the initial cleavage of mucin chains than the predicted periplasmic enzymes. We plan to test some of the other Gh16 enzymes against SI mucins in the future to assess differences in their specificity, but considering these will likely be subtle and not change the main findings of this work, we felt these experiments were beyond the scope of this paper.

Products were treated with an exo-acting hydrolyse.

Comment: It's unclear why they do this experiment and what is the conclusion. Everything they "conclude" is "likely" or "possibly".

The exo-acting enzymes in Fig2b were used to remove specific sugar decorations or terminal sugars from the GH16 products to aid in their structural characterisation. The data and description has now been moved to the Supplementary Information and Supp Fig. 8. The figure and text have been modified to make them clearer and more comprehensive.

Furthermore, they have already investigated the structure via MS/MS...

Fig. 2B: Please clarify what is meant by the red star?

The red asterisk was clarified in the legend originally to highlight the disappearance of particular peaks. These have now been split into three different colours, clarified in the legend, and an explanation added to the main text to make this clearer for the reader. This section is now in the supplementary information under the title 'Exo-glycosidase characterisation of GH16 enzyme products released from SI mucin'.

The MS/MS data can provide a lot of the structural data, but, for example, cannot distinguish between GalNAc and GlcNAc due to them having the same MW or the difference between types of linkages. To identify the α -GalNAc blood group structures, for example, we used a specific exo-acting enzyme. We think the combination of these GH16 enzymes, MS/MS data, and specific exo-acting enzymes could be useful in analysing O-glycan compositions in more detail in healthy and diseased samples (i.e. biomarkers).

Next, they tested the capability of the 9 enzymes to degrade keratan sulfate, which is structurally similar to mucin O-glycans. They detect that the 9 enzymes are active against keratan sulfates and that the enzymes can tolerate "a significant number of sulfates".

Comment: Why is this experiment done? What about the sulfates in the mucins they used in previous experiments?

This experiment was done for a number of reasons.

1. The substrate is structurally similar, but with very little fucose along the chains and more sulfate. Therefore, analysis of activity against this substrate shed light on how much sulfate could be accommodated.
2. Keratan sulfate is likely to be present in the gut from sloughed off epithelial cells and also dietary sources. Albeit, in amounts that aren't really known yet. Currently, we are not able to obtain enough to test bacterial growth, so by carrying out the assays we hoped highlight that this substrate may be used by members of the microbiota as a nutrient source.

Certain information has now been pulled from the Supplementary information to stress these points. Lines 207 onwards.

Next, they used a range of defined oligosaccharides to assess the substrate specificity of the 9 enzymes

Comment: Data and text are sound. However, a combining or concluding main figure would be nice for the main manuscript

We agree with the reviewer and I have generated Figure 4 to summarise the specificity of the different GH16 enzymes for the defined oligosaccharides.

Next, they tested the hydrolysis activity of the 9 enzymes towards previously published validated GH16 substrates

Comment: They could not really detect any hydrolysis of these substrates (besides some exceptions, which are discussed in the SM)

But: what is the conclusion? GH16 enzymes with specificity for mucins? What's the difference and the final conclusion?

The specificity of GH16 enzymes characterised here for mucin is unexpected as most GH16s characterised to date target marine and plant beta-glucans and -galactans. However, it is possible that mucinase activity is more common than we think and that previous studies on GH16 enzymes have just not tested mucins as a potential substrate. Therefore, to test this possibility we screen our enzymes against all previously determined GH16 substrates. In most cases the mucin active GH16s were inactive against plant and marine polysaccharides, confirming their specificity for mucin glycans. In the few cases where some activity against non-mucin glycans is observed, we have been able to link this to the structural data and explain why this might be. The points have been emphasised throughout now.

Overall, this paragraph is not really clear and comprehensive. The data of Fig. 2A are nice. However, it should be made clear what the purpose of Fig. 2B is. Why this is done and what are the conclusions of the "story". The text should mention that the red asterisks refer to the disappeared peaks.

As described in answer to an earlier question regarding Fig 2b, the experiment was done to characterise the GH16 products further using exo-acting enzymes with known specificity. We agree with the reviewer that this is not essential for the main story and have the data has now been moved to Supplementary information under the title 'Exo-glycosidase characterisation of GH16 enzyme products released from SI mucin' and the text expanded on to make the rationale for the experiments and conclusions clearer and more comprehensive.

Crystal structures:

Comment: Perhaps here they should clearly mention that this is done to elucidate the structural basis for mucin-specificity?

To highlight this point the first sentence of the structure section now reads: 'To investigate the structural basis for O-glycan specificity displayed by the GH16 enzymes, we attempted to obtain crystal structures of these enzymes in complex with substrate.'

They analyzed the apo-structures of BC2680, BC2680E143Q, BC3717 and AM0724.

Comment: Why were they chosen? Why the mutant?

The first paragraph of the structure section of the results has been changed to make this much clearer. We did aim to get structures for all enzymes and the ones in the manuscript are the successful ones.

Additionally, BF4060 and BC2680E143Q together with a trisaccharide.

Comment: What is the purpose of this? Why now adding BF4060? Why not comparing BC2680 with its mutant? Is this a catalytic mutant? Why not only perform co-crystallization of the holo-enzyme together with the substrate? Where does the trisaccharide come from? Was it added to the enzyme in its trisaccharide form or is it the product of an enzymatic reaction by these enzymes?

All incubations were done with TriLacNAc with the aim of obtaining an enzyme-substrate complex. Unfortunately, despite mutagenesis of one of the catalytic residues, the mutant retained enough activity for substrate hydrolysis to occur in crystal, hence a product rather than substrate complex was obtained. Again, the intro to the structure results section has been changed and more detailed added to clarify this point.

Activity against human mucins:

Comment: It should be mentioned in the text how the "human mucin samples" were generated and processed.

The main text section and methods have been expanded to address this.

They incubated the human mucin samples with AM0724 (why only this one?) and sialidase. Products were then labeled and identified.

These experiments were some of the last to be completed, so by this point we knew that Amuc0724 has a relatively broad activity – it is the best enzyme at accommodating blood group structures, for example, so it was used to maximise the types of O-glycans removed. This has now been added to the text.

They conclude that "enzymes could be used to characterise the glycan structures present in mucins from different disease states"

Comment: This conclusion is an overinterpretation and not supported by the available data. Is there a real difference between different disease states? There are no samples from healthy controls and only 2 samples from UC patients. The sample choice (uninflamed ileum and inflamed colon from Patient 1; inflamed ileum, uninflamed ileum and uninflamed colon) seems to be very random (and n=1 for each condition). The overall conclusion "showed different patterns of oligosaccharides released depending on the disease and tissue type" is not supported by the data. There are too many variables (individual, disease type, inflammation state, tissue sample, ...) and such a conclusion cannot be made by analyzing samples from only 1-2 individuals with a random sample selection. I am not certain about showing these data with definitive conclusions. I suggest to beef up the cohort size or tone down the conclusion based on the human data (including data on the CRC cell lines).

See answer to reviewers first comment above.

Reviewer #2 (Remarks to the Author):

This manuscript by Crouch et al. on the ability of O-glycanases from human gut microbiota members to digest mucin O-glycans via an endo-acting mechanism is well-supported by analytical and X-ray crystallographic studies. Some additional experiments and discussion are suggested to enhance the impact of this manuscript.

We thank the reviewer for their supportive comments about our work.

Major claims: (1) Gut bacteria access mucin O-glycans in the human intestine as a nutrient source. (2) Instead of acting as exoglycosidases, the enzymes secreted by these gut bacteria act via an endoglycosidase mechanism. (3) Understanding this mechanism is key to understanding human gut

microbiota interactions. (4) Recombinant O-glycanases can be useful tools to explore changes in the O-glycome of IBD and colorectal cancer.

Evaluation: There is significant interest in the human gut microbiome, and this manuscript describes detailed biochemical and structural evidence that glycosidases of the GH16 family from *Bacteroides* and *Akkermansia* act by digesting mucin glycans from porcine intestine via an endo mechanism instead of exo.

Biochemical evidence in the form of glycan LCMS is adequate, although the authors are suggested to further simplify Figure 2 for the non-glycoscientist.

We have put Fig 2b in the Supplementary Info and expanded the explanation for why the experiment was carried out and description of the main findings.

In particular, it is unclear why it is significant to group the enzymes based on products of more or less than four sugars (seems arbitrary).

This was initially because we'd thought it would help to see that the activities of the different enzymes broadly split between those that produce larger oligosaccharide products from mucin vs shorter products. But we agree this was somewhat arbitrary and we have removed it.

It would also be helpful to compare the structures of human versus porcine intestinal mucin to provide relevance to the study,

We agree this would be a very interesting comparison. It is currently unknown what a "normal" O-glycan composition is within a particular species or for a particular mucin type within that species, for example. Part of the significance of this paper is that the type of comparison you suggest is now more feasible with these enzymes and we have plans to produce this type of data in the future.

as well as gain a biochemical understanding of the expression levels of the enzymes describes in the bacteria specified (only previously reported transcriptomic dataset is referenced).

Does the reviewer mean quantitative proteomic data? Although all of the *Bacteroides* datasets we used were from transcriptomics studies, two of the datasets we used for *Akkermansia* were from proteomics studies (Sup Fig 3e,f) and these identified GH16 enzymes some of the most highly expressed proteins during growth on mucin. Thus while quantitative proteomics for the *Bacteroides* spp certainly be a useful complement to the transcriptomics data, we did not think this was essential to our study as it wouldn't change the main findings.

The cloning and expression of the recombinant proteins should also be described (right now the reader is referred to reference 25).

These details are now included in the Methods.

Enzymes used in this study were produced in an *E. coli* mutant, but how (if any at all) abundantly are these enzymes expressed, and how abundant are the requisite sialidases?

We were unsure whether the reviewer meant the level of expression in the native species or the expression we obtain in *E. coli*, so we answered both questions.

Native species

As detailed above the GH16s from *Akkermansia* have been shown to be highly expressed by proteomics, but as we haven't done quantitative proteomics for the *Bacteroides* spp, we can't be sure as to the protein levels of the *Bacteroides* enzymes. However, as the genes encoding the GH16s are some of the most upregulated in the *Bacteroides* spp. during growth on mucin we assume the protein levels are also high.

In *E. coli*

We are fortunate that enzymes from *Bacteroides* spp. generally express well and we are getting the same luck for *A. muciniphila* enzymes. Here is one example prep per enzyme of the concentrations obtained from 1 litre of cells (yields from different batches will vary slightly, but are usually quite reproducible):

Enzyme	mgs per litre of cells	Example concentration of this after metal ion affinity chromatography
--------	------------------------	---

		step and concentration (volume under 1 ml in μM)
BT2824	1.44	35.3
BC3717	0.58	26.0
BC02679	10.0	331
BC02680	0.34	11
BF4058	1.42	46.8
BF4060	0.75	24
Amuc0875	33.7	956
Amuc0724	15.8	468
Amuc2108	1.81	50.8
BT0455	0.3	5.1

Are these enzymes also active on N-glycans with extended poly-LacNAc chains?

Yes. We have shown that the Gh16 enzymes are active against the polyLacNAc structures found within N-linked Keratan sulfate Type I (Supp Fig 15).

N-glycans are also prominent members of the gut glycome. Is there specificity at the protein level as to which mucin glycoproteins are degraded by these enzymes (doesn't seem to, but explicit mention of this point would be helpful).

We're not sure what the reviewer is asking here. Are you wondering if the Gh16 enzymes will target only the O-glycan polyLacNAcs or potentially N-glycan structures as well? If so we think both as the KS data shows. Whether the oligosaccharides products we observed from the mucins were all from O-glycan is not clear, but it should be noted that PNGase treatment of the mucins used revealed only very low levels of N-glycan.

Can the SusC/D outer membrane complex be genetically manipulated to show that there is indeed uptake of the broken down glycan byproducts that could then be used for nutrition? More sophisticated experiments that more closely mimic the nature of the human gut microbiome would greatly enhance this manuscript.

It is not clear exactly what experiments the reviewer is suggesting here. Of the species we studied only *B. theta* can currently be genetically manipulated. It should therefore be possible to knock out the susCD genes from the BT2824-GH16 PUL and assess growth of the mutant strain on mucin, but it is not clear this would tell us about import of the GH16 products. However, we have attempted to delete the Bt2824 gene to further understand the enzymes role in mucin breakdown, but were unsuccessful despite multiple attempts in our lab as well as a collaborators.

However, to provide experimental evidence of Gh16 activity at the cell surface we have now included data showing endo activity on the surface of metabolically inactive cells against TriLacNAc that produces identical products to the recombinant GH16 mucinases.

Is mucosal viscosity a factor in the activity?

This is possible as there are differences in viscosity between the upper layer of colonic mucus (where gut microbes colonise) vs the lower sterile layer or between colonic and SI mucin so it is something we would like to explore, but have not done so yet as these experiments are not easy.

What about pH? The pH in the small intestine is ~ pH 6 and ~7.4 in the terminal ileum.

This is a good point and something else we would like to explore, but with the lab out of action at the time being this is not possible. In all cases the enzymes were assayed at pH 7.0.

Overall, these findings are of specific interest to the glycoscience-gut microbiome community, and the conclusions are original, but would be further strengthened by the suggested experiments. There is additional impact in adding information to carbohydrate-active enzymes.

While the authors' experimental conclusions are well supported, subjectively, I find a large disconnect in this small detail (exo vs endo mechanism) and it being incredibly significant and key to understanding gut-microbe interactions. It is not clear that this is a highly debated topic that is of general widespread interest to the wider community and that it is a key step to influence thinking in the field (although a better discussion of these specific topics could influence my evaluation).

The importance of gut microbes in host health (and disease) is well established, and so the nutrient sources these bacteria they access and how they modify host glycans is critical to understanding that

relationship. The transition from a model for mucin degradation that involves only exo-acting GHs to one where many species are using endo-acting enzymes will we feel raises interesting questions relating to this process including how gut microbes potentially share resources and how this may be important for community stability. The discovery of endo-acting enzymes also bridges the gap between the exo-acting GHs and the role of glycopeptidases in mucin breakdown. Furthermore, the phylogenetic analysis carried out indicated that endo-acting O-glycanases are prevalent in pathogens as well as commensals, which could provide greater understanding about the survival, mechanism of invasion and the pathogen-host relationship etc. O-glycan fragments have recently been shown to attenuate virulence in *P. aeruginosa*, for example, so there is likely a connection between commensals, mucin and keeping pathogens at bay. Hopefully the reviewer agrees that these points are of interest and important and are covered adequately in the discussion.

In my opinion, there is greater impact on the utility of these enzymes as tools to determine changes in patient samples, but there is not adequate data here for comparison (healthy samples are limited). We agree the use of the enzymes as tools is potentially very useful but that the data we present is very preliminary. We have tried to clarify this point in the revised MS and in the answer to the first comment of Reviewer #1 above.

Again, figure 4 can be better represented as digested data with bar graphs and statistical analysis instead of a series of chromatograms.

We feel that further analysis of the human samples would not add anything to the main finding, which was that the GH16 enzymes release oligosaccharide products from human mucin samples. However, the figure has now been altered and some of the data moved to supplementary info in an effort to make this point clearer.

Statistical analysis: Although the chromatograms used for glycan analysis are included and the description of methods used to generate them are satisfactory, there are limited mentions of any statistical analysis used. Instead of the chromatograms, bar graphs depicting the major conclusions replete with statistical measures of reproducibility are suggested.

The number of samples used is insufficient to use statistical analysis.

Reviewer #3 (Remarks to the Author):

This manuscript that reports a substantial volume of work on the identification and detailed characterization of the first endo-acting mucinases from a number of species of prominent human gut bacteria spanning two distinct phyla. Mucin samples from human were also shown to be degraded by one of the characterized enzymes. These endo-mucinases represent the first reported of such specificity in the large and well-studied glycoside hydrolase family 16, making this work of great interest to the carbohydrate enzyme community as well as to the gut microbiota community.

We thank the reviewer for their supportive comments about our work.

Regrettably, the presentation of the manuscript leaves much to be desired, and in many instances a clear lack of attention to detail is evident. It would seem that that the numerous co-authors have not carefully proofread the manuscript, which is burdensome to reviewers and regrettable considering the quality of the journal the authors are targeting. It is incumbent upon the corresponding authors to ensure that this is done properly.

We regret the reviewer found the presentation lacking and although this was not something mentioned by the other reviewers we have tried to address the comments as suggested and hope that the revised MS is now clearer.

The following comments must be addressed. These are presented generally in order of appearance in the documents.

Main text

Typographical/grammatical/sentence-structure problems must be addressed, for example on lines: 57, 58, 106, 116, 182, 278, 321, 354, 372, 378, 441, 445, 465, 499, 520. Note that this does not comprise an exhaustive list and the manuscript must be checked carefully throughout.

Corrected and we have checked the MS for other errors.

The sections titled “The GH16 enzymes are endo-acting mucinases” and “Crystal structures of O-glycan active GH16 family members” are big sections that present the bulk of the data; their content should be divided into smaller subsections with informative headers to improve readability.
We agree and have split these sections to make clearer.

It would be more logical for the section titled “Phylogenetic analysis of the GH16 O-glycanases” to be moved up, directly after the section “Identification of GH16 enzymes expressed during growth on mucin”. Also, this way it is consistent with the figure order and supplementary discussion order. Content that is best presented after the enzymology and structural biology sections can go in the discussion.

We have rearranged sections, tried to make everything more readable and flow better. The phylogenetic section has been moved earlier in the paper.

For the section titled “The GH16 O-glycanases are active against human mucins from healthy and diseased samples”, more interpretation of the results is required. The first paragraph describes the sample source, the second paragraph is discussion material (should really be moved to the discussion section), and only the third paragraph (lines 327-333) describes the results. Maybe elaborate a bit on the pattern of oligosaccharides released as a function of disease state.

We agree this would be ideal, but we are reluctant to elaborate too much on the details of oligosaccharides produced due to the small sample size. However, we have changed this section to highlight the rationale for these experiments more clearly. Please see the response to reviewer #1's first comment for details on this issue.

In light of the possibility of use as a biomarker, are there perhaps any particular unique or uniquely enriched oligosaccharides corresponding to a disease?

This cannot currently be determined.

Line 95: Define “Sus” and explain.

This has been defined. Line 84 onwards.

Line 109-112: with regard to ref. 29, it would seem that the linkage specificity of the enzymes in the present study is not entirely unprecedented/surprising. Here, and in the corresponding structural biology sections, the structural similarities and differences with regard to carbohydrate recognition should be explicitly described, including through the use of structural superpositions.

The linkage specificity may be unsurprising for this family, but the targeting of polyLacNAc structures in mucins is unexpected. These enzymes are the first to accommodate an alternating glucose and galactose polymer in addition to pockets/space for decorations that are highly variable.

We have discussed these points as well as the structural similarities and differences with regard to carbohydrate recognition between the enzymes characterised here and previously described GH16s in the main text and with further details in the supplementary info structural sections.

The initial approach to analysing our structural data involved evaluating the already available structures and trends or conclusions that had been highlighted in the literature (main text summary and Supplementary in detail). We highlight the variety of substrates that this enzyme family can deal with by showing the cleft shape in enzymes with different specificities from two different angles and the *C. perfringens* exo-acting GH16 structure is shown alongside these (Supplementary Figure 18). From there, the overall shape of the binding cleft and each subsite for our new structures were considered separately from the perspective of glucanases, laminarinases, agarases, porphyranases, carrageenases, xyloglucanases and the exo-acting GH16. For instance, details of the -1 subsites of enzymes with different specificities were systematically looked at in terms of shape, side chains and the loops/fingers where these residues came from for enzymes displaying all the different activities (detailed in Supp info, ‘Accommodation of a glucose or galactose at the -1 subsite in GH16 family members.’).

We condensed our observations down to four main points for a reader and these are supported by more details in the supplementary info and figures throughout (e.g. Fig 2e). We did attempt to produce figures of structural overlays, but felt these figures were quite confusing and the information could be communicated in a more understandable way. For example, we frequently overlay other substrates/products into the active sites of the O-glycanases to demonstrate key structural features (e.g. Supp fig 20).

Line 114: What does “sequence diverse” mean in this context? Presumably, a subfamily would have comparatively low diversity compared with the larger GH16 family.
This has been clarified in the text by using % identity.

Line 128: PGM = porcine gastric mucin.
Corrected.

Line 144: Please elaborate on the structural basis for this specificity, especially with regard to the cleft-like active-site of these enzymes (see also comment on Lines 109-112 above).
See above for detailed response.

Line 154: Where is the data that shows this? I think it is supposed to be in Supplemental Figure 13 but is missing (see below).
The data showing lower activity for this enzyme is in Supplemental Figs 7 and 11 – new figure numbering. Figure 3 (originally Fig 2A) also shows relatively low peaks being produced, indicative of low activity. The Supp figure you mention – substrate depletion experiments were not possible for this enzyme, because the activity was too low.

Line 155: Was there a particular reason for the selection of BT4058 for detailed product analysis?
BF4058 is one of the enzymes predicted to be surface located and we were trying to assess what could be happening on the cell surface. Analysing all the enzymes against all the mucin samples will be something we would like to look at in the future, but would be a significant undertaking and we felt was beyond the scope of this paper.

Line 212-217: There are six structures described here but only stats for four of the structures in sup table 5.

We apologise to the editor and reviewers. A working version of the table was submitted rather than our final version. The up to date table has been submitted with the revised MS.

Line 217: Sup figs 17-19 not 15-17.
Corrected.

Line 214-218: Privateer [Nature Structural & Molecular Biology volume 22, pages 833–834 (2015)] must be used to validate these sugar models and the resulting output reported in a supplemental table. (See also identical comment in SI section below).

We have carried out this analysis, created the new supplemental table 7 and supp Fig 17 and modified the text in the main, supplemental and methods sections.

Line 276: Meaning unclear (misplaced modifier).
Corrected.

Line 271: Sup fig 16 not 18.
Corrected.

Line 281: Run-on sentence/poor grammar.
Corrected.

Line 327: Was there a particular reason for the selection of Amuc_0724 for assays with human mucin samples?

Amuc0724 has a broad activity – for example it is the best enzyme at accommodating blood group structures, so it was used to maximise the types of O-glycans removed. This info has now been added to the text. Line 388.

Line 435: Which is it, soak or co-crystallized? If both were tried, which led to the structure presented?
All our screens were co-crystallisation and we have modified the main text to make this clearer.

Line 456-457 and rest of methods section: the preferred abbreviation for milliliter is mL not ml (same for microliter; uL not ul)

I think this is the journal specifications rather than personal preference.

Figure 2: It should be clearly stated what the control is. Just sialidase?

Corrected.

Figure 3: Baccac_02680 structure should be labelled as the E143Q mutant.

Corrected.

In panel b, why are the catalytic residues colored red in Bf4060 but not in Baccac_02680? Be consistent. Similar issue in sup fig 19.

Corrected.

Check caption carefully – Beta1,3-linkage, “insert number here”,

Corrected.

the supplementary figure you want to refer readers to is 19 not 17.

Corrected.

Supplemental Information

Typographical/grammatical/sentence-structure problems must be addressed on lines: ~~110, 166, 226, 265, 301, 504~~

Corrected.

Line 190-198: Please cut down on these texts which are redundant with lines 193-205 from the main text. The supplemental discussion should complement the main text instead of rephrasing the same thing.

While we agree with the reviewer that we don't want to repeat what has already been said, we feel that this introductory sentence is important to put the results in context. We have cut down redundant text where possible.

Line 209-213: These texts are likewise redundant with main text lines 181-192.

See above.

Line 30-527: Carefully go through every section and remove all instances of redundancy with the main text.

We have done this.

Line 316-323: I'm curious how convincing the electron density that supports skew boat conformation is at 2.7 Å resolution for Baccac_02680. There is not a single figure with electron density maps; 2Fo-Fc maps at a minimum, and ideally composite omit maps, should be incorporated into figure.

A new figure has been inserted to show the omit maps of the ligands in detail. (SF17)

Line 316-323: Privateer [Nature Structural & Molecular Biology volume 22, pages 833–834 (2015)] must be used to validate these sugar models and the resulting output reported in a supplemental table.

As above.

Line 321-323: So the density was too poor to define the ligand in BF4060 and it was built anyway? Same comments as previous: please show appropriate maps and run Privateer validation.

We have run the validation and re-written this section.

Line 345-346: What does “different evolutionary origins” mean here? Presumably all enzymes in this subfamily have a common ancestor (hence the ability to construct a meaningful phylogeny)?

This section has been reworded in parts to make the meaning clearer.

Table 3: The “prefixes used for brevity” are really superfluous and should be replaced with the “true locus tag prefixes” throughout. In most cases they are the same, or do not result in the remove of many characters.

The manuscript now contains full locus tags throughout, unless otherwise states.

Is this just laziness regarding figure editing?

Table 5: Completely missing all statistics for apo Baccac_02680 and Baccac_02680 E143Q. For Amuc_0724 and BF4060 were there really zero water molecules? Missing Ramachandran stats for BF4060. Table caption: "reflections that were excluded from.....".

We apologise to the editor and reviewers. A draft version of the table was submitted rather than the final version. The up to date table has been submitted with the revised MS. The data for Amuc0724 and BF460 at resolution cut off of 2.7 and 3.3 Å, respectively, does not allow modelling of single water molecules.

Table 6: What is L404 ligand?

This has been removed.

Figure 4: Please make the distinction between e) and f) clearer in the figure caption because they are both mixed-linkage glucan. Xyloglucan in i) is inaccurate.

The figure has been changed to emphasise the differences between the beta-glucans. The xyloglucan is correct but the linkages have been annotated on the panel rather than following the key as stated in the legend.

Figure 5: Label figure panels

Does the reviewer mean Supp Fig. 5? This figure has now been deleted as a consequence of rearranging the order of the data.

Figure 6 and 7: I am unable to find a look-up table as described in the legend.

We apologise for this error. The table is no longer required in the revised MS.

Regardless, most of the branch labels are impossible to read. Overall, these trees should be presented with an alternate layout that avoids overlap; a rectangular format would be far preferred, using collapsed branches where advantageous. A properly organised and displayed tree would not require all of the accessory text annotations and arrows.

Figure 6 and 7: Insufficient information is provided regarding the algorithms used for sequence alignment and phylogeny estimation. Presently, there is no information on the statistical significance of any of the "Branches" described. At a minimum, a Maximum Likelihood phylogenies should be presented, using at least 100 bootstraps, and significant bootstrap values must be indicated on the published trees. A suitable outgroup should be used to root the trees; in the case of Figure 7, presumably one or more related GH16 subfamilies could be used.

Further information has been provided in the Materials and Methods about how the trees were produced.

Both figures are now in the requested rectangular format, so labels can now be read. Supp Fig. 6 has had the numbers replaced with Genbank numbers/uniprot codes/locus tags. In Supp. Fig. 7, some of the accessory annotations have been kept, but simplified. We believe this allows the reader to easily interpret the different areas of the tree without having to go and look up the different components individually.

Regarding bootstrapping, under the current Covid lockdown, we do not have access to the computing power to produce trees with this many sequences using 100 bootstraps. This was initially attempted prior to lockdown with the Supp Fig. 6 sequences and the program was still running after 48 hours.

The trees were produced to put the new activities of the O-glycanases into the context of 1) the currently described activities and 2) subfamily 3. A detailed phylogenetic analysis has recently been completed for this family, resulting in grouping the sequences into subfamilies (Viborg et al 2019; PMID: 31501245). We agree it would be preferable to make more robust trees if possible, but hope the reviewer agrees in this instance the data is valid for the conclusion we are drawing.

The second tree (originally Supp Fig 7) already included outgroups in the form of two sequences from *Bacteroides* spp. from subfamily 12 and 16 with known activities.

This description of the different "branches" has been cut.

Figure 6 and 7: Please provide the input multiple sequence alignment files in FASTA format and trees in a suitable format (e.g. Newick, PhyloXML) as supplemental information files to allow reader direct access to the underlying data.

These are now included as text files.

Figure 8: The last sentence of the caption is slightly confusing. Please make it clear that BT0455 is the sialidase and GH95 is the fucosidase.

Corrected.

Figure 11: Is species 14 correctly depicted? There is a stray GlcNAc on the end of a row of sulfations. Yes this is correct. The MS/MS data was not enough to tell where this sugar was in the structure, so could be a branching sugar.

Figure 13: Panel E is missing and the panel labels on the figure are not consistent with their description in the figure caption. Is low-activity Amuc_0875 the missing data here?

The legend and panel labels have been corrected.

Figure 15: The lanes are not labeled.

Corrected.

Figure 16: μM enzyme not μm , it is a unit of concentration not length.

Typo corrected.

Figure 16: Negative results determined only by TLC analysis is not particularly compelling, especially given weak staining efficiency (high limit of detection) and some streaking at Rf values consistent with oligosaccharides is observed. Supporting data using a standard reducing sugar activity assay must be provided to quantify relative specific activities.

We agree that TLC is not the most sensitive technique for detecting sugars. However, we typically spot 3 μl of a 1mM sample out, which works out $\sim 0.3\text{-}1.5\ \mu\text{g}$ depending whether it's a monosaccharide or larger glycan and is clearly detectable by TLC (demonstrated throughout the manuscript). Therefore, we can be sure to detect very low amounts of product(s). For the polysaccharides in this Fig, the substrates were used at a final conc of 10 mg/ml and 3 μl was spotted (30 μg). Therefore, if no activity can be seen by TLC at the high substrate and enzyme concentrations used over the long incubation period then we would propose based on ours and other previous studies that this can confidently be considered as no, or at least very low, activity.

While reducing sugar assays could potentially show some weak activity against some of the substrates tested that does not show up by TLC, we do not feel this would change our interpretation of the data. Furthermore, reducing sugar assays against some of these glycans can be technically challenging and very time consuming if they require pre-reduction to reduce the background reading even before carrying out kinetics of all nine enzymes against eighteen substrates.

Nevertheless, we have tried to measure the activity of the GH16 O-glycanases against mucins using a reducing sugar assay. However, we found high background and were unable to remove this despite trying a variety of different methods to reduce background prior to assaying.

Figure 17: It would be helpful if the relevant loops are labeled with their finger number. Also, consider moving this figure to after 18 and 19 as its content is first referred to from the text after first reference to 18 and 19.

Labels have been added to the loops as requested and the figures re-ordered to suit the text.

End comments

REVIEWERS' COMMENTS:

Reviewer #2 (Remarks to the Author):

This revised manuscript by Crouch et al. is significantly better than the original submission. The significance and discussion is now much better conveyed, and the conclusions are appropriate for the study. The authors have also sufficiently addressed the reviewers' comments and suggestions.

Reviewer #3 (Remarks to the Author):

In this revision, the authors have effectively addressed nearly all of my previous comments, resulting in a much improved manuscript. In particular, the logical flow of the text is significantly better and easier to follow, and the correction of numerous errors and omissions in the data presented greatly improves confidence in the analysis. I only have a few minor comments on this revision:

1. Regarding my original comment on the multiple sequence analysis and phylogeny, "Please provide the input multiple sequence alignment files in FASTA format and trees in a suitable format (e.g. Newick, PhyloXML) as supplemental information files to allow reader direct access to the underlying data."

- I had requested that the authors provide their *alignment* files used as input for the phylogenetic analysis (i.e. the CLUSTAL output, which should contain alignment gaps), but it seems that the two SI files provided with this revision are the input files for the CLUSTAL alignment. The point here was to allow reader to examine the alignment quality, on which the phylogeny is dependent. Please provide the CLUSTAL output files in FASTA format (as a simple text file, if possible, not a Word file).

- I do not see tree files included among the SI files in the revision. Please provide them.

- Lastly, bootstrap values should be indicated on the published trees, at least above a certain statistical value, e.g. 50%. I appreciate that the authors were unable to run the usual 100 ML bootstraps due to computing limitations, but some indication of branch robustness should be provided in the figures.

2. lines 111-114: Given that a number of GH16 members specifically from *Bacteroides* species have been characterized with porphyranase, agarase, beta1,3 glucanase and mixed-linkage glucanase activities, a statement to this effect and citation of the primary literature would be helpful for context (PMIDs 23150581, 29535379, 32265336, 29020628).

3. lines 332-334 (also to provide context for line 444): Please provide supporting primary references here, especially to functionally and structural characterized GH16 subfamily 3 enzymes (http://www.cazy.org/GH16_characterized.html). Also, I believe line 334 should say "mixed linkage glucan".

4. line 350: Do the authors wish to refer to the +1 subsite, specifically?

5. line 356: "its" (not "it's" = it is)

6. line 390: "was"

6. lines 390-393: While I appreciate that the authors are being justifiably cautious here in avoiding prescribing specific biomarkers, it would seem to be valuable to future researchers to quantify the available HPLC data in a table. The chromatograms in Fig. 7 show some very obvious differences

between samples, e.g. the predominance of peaks from (2) in panel a, varying amounts of (13) across all samples, etc. Using the existing HPLC data, please provide an SI table of peak areas for all glycans identified. Ditto for the data in SI figure 21.

7. Figures 1 and 3: Please include the linkage key here also.

8. SI figures 17 and 18. The last sentence of the corresponding legends appears to be incomplete or grammatically incorrect.

End of comments.

Dear Reviewers,

Thank you for taking the time to look at our work again, it is much appreciated, especially under the current conditions. We also appreciate your final comments and suggestions and hope the responses are satisfactory.

Best wishes
Lucy and Dave

REVIEWERS' COMMENTS:

Reviewer #2 (Remarks to the Author):

This revised manuscript by Crouch et al. is significantly better than the original submission. The significance and discussion is now much better conveyed, and the conclusions are appropriate for the study. The authors have also sufficiently addressed the reviewers' comments and suggestions.

We thank the reviewer for their supportive comments about our work.

Reviewer #3 (Remarks to the Author):

In this revision, the authors have effectively addressed nearly all of my previous comments, resulting in a much improved manuscript. In particular, the logical flow of the text is significantly better and easier to follow, and the correction of numerous errors and omissions in the data presented greatly improves confidence in the analysis. I only have a few minor comments on this revision:

We thank the reviewer for their supportive comments about our work.

1. Regarding my original comment on the multiple sequence analysis and phylogeny, "Please provide the input multiple sequence alignment files in FASTA format and trees in a suitable format (e.g. Newick, PhyloXML) as supplemental information files to allow reader direct access to the underlying data."

- I had requested that the authors provide their *alignment* files used as input for the phylogenetic analysis (i.e. the CLUSTAL output, which should contain alignment gaps), but it seems that the two SI files provided with this revision are the input files for the CLUSTAL alignment. The point here was to allow reader to examine the alignment quality, on which the phylogeny is dependent. Please provide the CLUSTAL output files in FASTA format (as a simple text file, if possible, not a Word file).

Apologies to the reviewer. The alignment files are now included.

- I do not see tree files included among the SI files in the revision. Please provide them.
The tree files are now in the Source Data Folder and have been saved in every possible way.

- Lastly, bootstrap values should be indicated on the published trees, at least above a certain statistical value, e.g. 50%. I appreciate that the authors were unable to run the usual 100 ML bootstraps due to computing limitations, but some indication of branch robustness should be provided in the figures.

The bootstrap values can be generated by using the files in the Source Data Folders. To add them on at this stage would be time consuming as the figures would have to be annotated by hand for a third time. We did look into this and feel the added information also makes an already crowded figure look worse. We appreciate that the reviewer wants as much accuracy as possible in the information displayed, but with these figures we just aim to provide some context for where the O-glycanases sit within 1) currently characterised GH16

family members and 2) other subfamily 3 members. Other groups with more experience could most likely produce a more accurate analysis, but we think it is not required for this manuscript.

2. lines 111-114: Given that a number of GH16 members specifically from Bacteroides species have been characterized with porphyranase, agarase, beta1,3 glucanase and mixed-linkage glucanase activities, a statement to this effect and citation of the primary literature would be helpful for context (PMIDs 23150581, 29535379, 32265336, 29020628). We agree with the reviewer that this information is a very nice addition and a sentence has been added.

3. lines 332-334 (also to provide context for line 444): Please provide supporting primary references here, especially to functionally and structural characterized GH16 subfamily 3 enzymes (http://www.cazy.org/GH16_characterized.html). We have now included a reference.

Also, I believe line 334 should say "mixed linkage glucan".
Thank you for that catch.

4. line 350: Do the authors wish to refer to the +1 subsite, specifically?
Yes, thank you, this has now been changed.

5. line 356: "its" (not "it's" = it is)
Corrected.

6. line 390: "was"
Corrected.

6. lines 390-393: While I appreciate that the authors are being justifiably cautious here in avoiding prescribing specific biomarkers, it would seem to be valuable to future researchers to quantify the available HPLC data in a table. The chromatograms in Fig. 7 show some very obvious differences between samples, e.g. the predominance of peaks from (2) in panel a, varying amounts of (13) across all samples, etc. Using the existing HPLC data, please provide an SI table of peak areas for all glycans identified. Ditto for the data in SI figure 21. For this type of HPLC it is not possible to use the areas under the peaks as reliable data. Although the ratios of different products are most likely a reflection of the true ratios, there is no direct evidence of this. The procainamide labelled samples go through labelling and clean-up processes and the different glycans may be recovered slightly differently, thus skewing the ratio. The analysis has to be binary – is it there or not. Once important biomarkers have been identified (by using exo-acting enzymes of particular specificities) it will be possible to start developing standards to be able to use ratios. Direct application of a sample onto the HPLC would solve this issue and allow us to start quantifying peaks, but this isn't possible with these samples currently as they are in very low concentrations and fluorescent labels are required.

We added these sentences at the end of the results section to improve the analysis:
A total of 22 different oligosaccharides were detected in the various samples analysed and the glycan profiles observed varied significantly between samples (Fig. 7 and Supplementary Fig. 21). Furthermore, multiple peaks have the same sugar composition, indicating the different peaks are due to variation in linkages between the sugars and that these can be differentiated using this approach (e.g. Fig. 7a, glycan 2). The variation in glycan profiles observed using the O-glycanase digestion suggest that these enzymes could be promising tools to discover disease-specific biomarkers.

7. Figures 1 and 3: Please include the linkage key here also.
Completed.

8. SI figures 17 and 18. The last sentence of the corresponding legends appears to be incomplete or grammatically incorrect.
These have now been corrected.

End of comments.